# Information-Geometric Adaptive Sampling for Graph Diffusion

**Yuhui Lu** [1]  **Wenjing Liu** [1]  **Kun Zhan** [1]

## Abstract

Standard diffusion models for graph generation typically rely on uniform time-stepping, an approach that overlooks the non-homogeneous dynamics of distributional evolution on complex manifolds. In this paper, we present an information-geometric framework that reinterprets the diffusion sampling trajectory as a parametric curve on a Riemannian manifold. Our key observation is that the Fisher-Rao metric provides a principled measure of the intrinsic distance. By analyzing this metric, we derive the Drift Variation Score (DVS), a geometry-aware indicator that quantifies the instantaneous rate of distributional change. Unlike prior heuristic-based adaptive samplers, our DVS solver enforces a constant informational speed on the statistical manifold, automatically maintaining a uniform rate of distributional change along the sampling trajectory. This equal arc-length strategy ensures that each discretization step contributes equally to the information speed. Theoretical analysis verifies that DVS characterizes the local stiffness of the sampling dynamics in the Fisher-Rao sense. Experimental results on molecule and social network generation show that DVS significantly improves structural fidelity and sampling efficiency.

## 1. Introduction

Graph generation is fundamental to a wide range of real-world applications, including molecular design (Bao et al., 2023), social network analysis (Grover et al., 2019), and knowledge graph construction (Zhu et al., 2019), where complex relational structures and combinatorial constraints pose significant challenges. Recently, diffusion-based generative models have emerged as a prominent framework for this task by characterizing complex graph distributions

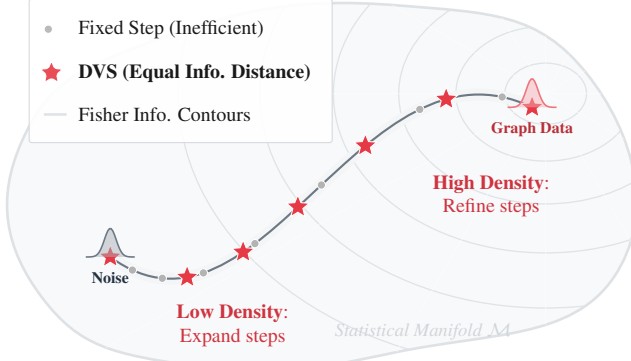

*Figure 1.* **Conceptual illustration of DVS-driven adaptive sampling on the statistical manifold** $\mathcal{M}$. We compare the standard fixed-step discretization (gray circles) with our DVS sampler (red stars). Fixed-step methods fail to account for the non-uniform dynamics of the sampling trajectory, leading to inconsistent informational progress. In contrast, our DVS sampler maintains a constant informational distance per step under the Fisher-Rao metric. By dynamically sensing the local geometry, it automatically **refines the step size** in high-information-density regimes to ensure numerical stability and **expands the step size** in stable, low-curvature regimes to accelerate the sampling process.

through continuous-time stochastic processes. Sampling from these models typically involves numerically integrating the reverse-time stochastic differential equation (SDE) via a predictor-corrector (PC) sampler (Song et al., 2021b; Song & Ermon, 2019). In this paradigm, standard solvers like Euler-Maruyama (Maruyama, 1955) or Heun (Karras et al., 2022) act as predictors on uniform time grids, followed by Langevin-based refinement to align the generated samples with the learned score field.

Although fixed-step discretization has become standard practice in diffusion sampling, it implicitly assumes that the reverse-time dynamics evolve at a uniform rate throughout the time horizon. However, the reverse-time SDE in graph diffusion models exhibits highly non-uniform behavior across different time regions (Lu et al., 2022). The high-noise regime is characterized by stable dynamics as the model delineates the macro-topology. Conversely, during the low-noise regime, the dynamics become stiff and exhibit rapid changes (Zhang & Chen, 2023), where small variations in time lead to pronounced differences in the drift. Under a fixed discretization scheme, such non-uniform

[1]School of Artificial Intelligence, Lanzhou University, Gansu, China. Correspondence to: Kun Zhan <kzhan@lzu.edu.cn>.

*Proceedings of the 43rd International Conference on Machine Learning*, Seoul, South Korea. PMLR 306, 2026. Copyright 2026 by the author(s).

dynamics result in inefficient allocation of computational resources (Ho et al., 2020; Kong & Ping, 2021): smooth regions are unnecessarily refined with small steps, while regions with rapidly varying dynamics are insufficiently resolved, potentially leading to significant numerical integration errors. For molecules, these issues are further amplified by discrete structural transitions and the desynchronized denoising rates of nodes and edges (Jo et al., 2022).

Several approaches (Bao et al., 2022; Lu et al., 2022; Kim & Ye, 2023) have been proposed to improve diffusion sampling by optimizing the distribution of time steps. Heuristic schedules, such as the quadratic schedule (Song et al., 2021a) and other power-law reparameterizations, attempt to manually concentrate more steps in sensitive time regions. While these approaches often offer gains, they are essentially static prescriptions that cannot adapt to the actual variability of different datasets or the specific stiffness of different model architectures. Alternatively, some strategies employ adaptive step-size control borrowed from numerical analysis, typically relying on local truncation error (Karras et al., 2022) estimated in the state space. However, these methods overlook the intrinsic geometry of the probabilistic path, failing to adaptively balance numerical fidelity against the varying stiffness of the statistical manifold.

To bridge this gap, we adopt an information-geometric perspective, treating the evolution of transition kernels induced by the reverse-time SDE as a parametric curve on a Riemannian statistical manifold (Amari, 2016; Girolami & Calderhead, 2011). In this view, the sampler's local behavior is governed by the intrinsic geometry of the evolving transition distributions. By equipping this curve with the Fisher-Rao metric (Liang et al., 2019), we derive a formal measure of local sensitivity, termed the *Drift Variation Score* (DVS), that quantifies the instantaneous rate of informational change.

This geometric signal motivates the DVS-*driven adaptive sampler*, which dynamically scales the step size to enforce a uniform rate of distributional change along the sampling trajectory, resulting in a constant informational speed on the statistical manifold. This mechanism effectively prevents large discretization errors in high-curvature regions where the drift varies rapidly, while accelerating the process in stable regimes. We provide an illustration of this mechanism in *Figure 1*, which highlights the contrast between the non-uniform progress of traditional schedules and the constant informational distance per step maintained by DVS.

We evaluate the DVS-driven adaptive sampler on various benchmarks including molecular and general graph datasets. Across these tasks, our method consistently outperforms uniform and quadratic-scheduled versions of Euler and Heun, achieving superior generation quality and stability while maintaining comparable or improved efficiency. Our main contributions can be summarized as follows:

- We establish a rigorous information-geometric perspective on graph diffusion sampling by formalizing the sequence of numerical transition kernels as a parametric curve on a Riemannian statistical manifold equipped with the canonical Fisher-Rao metric.

- We derive the DVS, a training-free and interpretable metric that explicitly quantifies the instantaneous rate of informational change. This score serves as a principled feedback signal to detect the local sensitivity and numerical stiffness of the reverse-time dynamics.

- We design a DVS-driven adaptive sampler that advances the reverse-time SDE with approximately constant information distance per step. The proposed method introduces negligible computational overhead and can be seamlessly integrated into existing graph diffusion frameworks.

Through comprehensive experiments on molecular and general graph datasets, we demonstrate that our DVS sampler consistently enhances structural fidelity and numerical stability, outperforming both standard fixed-step and heuristic non-uniform schedules.

## 2. Related Work

**Graph Diffusion Models**   Recent studies (Jo et al., 2022; Hoogeboom et al., 2022; Niu et al., 2020; Vignac et al., 2023; Xu et al., 2022; Qin et al., 2025) have successfully adapted diffusion-based modeling to graph domains by defining continuous-time stochastic differential equations (SDEs) over node and edge attributes. In these frameworks, a neural network is trained to approximate the reverse-time dynamics, enabling sample generation through the numerical solution of the corresponding reverse SDE (Anderson, 1982). These sampling tasks predominantly rely on fixed-step solvers, notably the first-order Euler-Maruyama (Maruyama, 1955) and second-order Heun (Ascher & Petzold, 1998) methods, that discretize the reverse-time dynamics uniformly, thereby implicitly assuming that equal time intervals correspond to comparable changes in the distribution. In practice, this assumption often breaks down along the reverse diffusion trajectory: near the low-noise regime close to the data distribution, statistical sensitivity increases sharply so that a fixed time step becomes too large and may lead to instability or loss of accuracy, whereas in high-noise regimes where the distribution evolves smoothly and statistical curvature is low, fixed-step discretization can be overly conservative, resulting in unnecessarily small steps and inefficient sampling. These limitations motivate our adaptive sampling strategy, which dynamically shrinks or enlarges step sizes according to the local geometry of the evolving distribution space.

**Ornstein-Uhlenbeck Bridges**   Recent advancements in

graph diffusion have increasingly focused on Ornstein-Uhlenbeck (OU) bridge (Jo et al., 2024) processes, which offer several theoretical advantages over standard SDE-based diffusions (Jo et al., 2022). Unlike conventional diffusion models that only approximately reach the prior at $T$, the OU bridge enforces an exact endpoint constraint, ensuring that the sampling trajectory terminates precisely at the target distribution and thereby mitigating distribution shift and preserving sensitive graph topologies. In addition, the mean-reverting property of the OU process (Uhlenbeck & Ornstein, 1930) introduces a stabilizing drift that continuously pulls the trajectory toward a predefined mean, preventing the generative process from wandering into invalid regions of the high-dimensional adjacency space and producing robust graph skeletons. As an instantiation of the *Schrödinger bridge* (De Bortoli et al., 2021; Chen et al., 2022; Liu et al., 2023), OU bridges yield entropy-optimal transport paths with lower inherent reverse-time curvature. Despite the theoretical potential for larger integration steps, existing models typically employ fixed, geometry-agnostic schedules. Consequently, OU bridges provide a robust foundation for exploring adaptive, geometry-aware step-size control.

**Sampling Strategies** To bypass the prohibitive cost of retraining large-scale models, recent research has shifted toward optimizing the sampling phase (Tong et al., 2025; Watson et al., 2022; Chen et al., 2024; Xue et al., 2024). A prominent direction involves optimizing discrete sampling schedules (Kahouli et al., 2024). Early heuristic approaches, most notably the quadratic schedule (Song et al., 2021a), attempt to improve fidelity by manually densifying steps in sensitive regimes via fixed power-law reparameterizations. More advanced techniques include AYS (Sabour et al., 2024), which employs error-aware time reparameterization based on numerical error propagation, and JYS (Park et al., 2025), which utilizes non-uniform jumps to skip uninformative intermediate states. Additionally, S++ (Yu & Zhan, 2025) mitigates reverse-starting bias and exposure bias through Langevin-based initialization alignment and score correction. Despite these advances, both heuristic approaches like the quadratic schedule and error-based ones like AYS remain largely geometry-agnostic. They rely on static prescriptions or state-space estimates rather than the intrinsic statistical geometry and instantaneous dynamics of the trajectory on the statistical manifold. Several recent works explore information-theoretic guidance for diffusion sampling, particularly via Fisher information. For instance, Song & Lai (2024) reweight the score function using the Cramér-Rao bound to modify the sampling dynamics. Complementary to this line of work, we instead focus on the discretization of the sampling trajectory and develop an adaptive sampler that dynamically adjusts step sizes in response to real-time information variation.

## 3. Methodology

### 3.1. Reverse-Time SDE and Sampling Dynamics

Diffusion-based generative models characterize the transformation between a simple prior and a complex data distribution. In this work, we define the generative process as a stochastic trajectory $\boldsymbol{x}_t \in \mathbb{R}^d$ over the time interval $t \in [0, T]$. This process is governed by a generative stochastic differential equation (SDE) that evolves from a tractable prior noise distribution at $t = 0$ toward the target data distribution at $t = T$:

$$\mathrm{d}\boldsymbol{x}_t = \boldsymbol{f}_t \, \mathrm{d}t + g_t \, \mathrm{d}\overline{\boldsymbol{w}}_t \tag{1}$$

where $\boldsymbol{f}_t$ is the reverse-time drift term, which is typically learned via denoising score matching, $g_t$ is the diffusion term, and $\overline{\boldsymbol{w}}$ represents a reverse-time Wiener process. For an infinitesimal time increment, the Euler-Maruyama discretization of Eq. (1) yields the update rule:

$$\boldsymbol{x}_{t+\Delta t} = \boldsymbol{x}_t + \boldsymbol{f}_t \, \Delta t + g_t \sqrt{\Delta t} \, \boldsymbol{\epsilon}, \quad \boldsymbol{\epsilon} \sim \mathcal{N}(\boldsymbol{0}, \boldsymbol{I}). \tag{2}$$

To characterize the local manifold structure, we consider the transition density $p$ induced by a small time increment $\mathrm{d}t$. At any state $\boldsymbol{x}_t$, this density can be viewed as a family of Gaussian distributions parameterized by the local drift vector $\boldsymbol{f}_t \in \mathbb{R}^D$:

$$p(\boldsymbol{x}_{t+\mathrm{d}t}|\boldsymbol{x}_t; \boldsymbol{f}_t) = \mathcal{N}(\boldsymbol{x}_t + \boldsymbol{f}_t\mathrm{d}t, \, g_t^2\mathrm{d}t \, \boldsymbol{I}). \tag{3}$$

The collection of these densities forms a Riemannian statistical manifold $\mathcal{M}$. The mapping $\boldsymbol{f}_t \mapsto p(\cdot; \boldsymbol{f}_t)$ defines the drift vector as a local coordinate system, endowing the set with a differentiable manifold structure (Amari, 2016). We equip this manifold with the Fisher-Rao metric (Liang et al., 2019) $\mathcal{I}(\boldsymbol{f}_t)$ as the Riemannian metric tensor. This choice is principled due to Chentsov's Theorem (Amari, 2016), which identifies the Fisher Information as the unique metric invariant under sufficient statistic transformations. From this information-geometric perspective, the numerical integration of the reverse-time SDE defines a parametric curve on $\mathcal{M}$, and the Riemannian line element $\mathrm{d}s^2$ is defined as the infinitesimal arc length under the Fisher-Rao metric, measuring the distance between neighboring transition distributions along the curve.

### 3.2. Fisher-Rao Metric Induced by the Drift Field

To specify the line element for the generative process, we treat the noise scale $g_t$ as locally constant, as its variation along the sampling trajectory is several orders of magnitude smaller than that of the drift field (see Appendix A). Consequently, the line element $\mathrm{d}s^2$ is primarily associated with infinitesimal changes in the drift parameter $\boldsymbol{f}_t$:

$$\mathrm{d}s^2 = \mathrm{d}\boldsymbol{f}_t^\top \mathcal{I}(\boldsymbol{f}_t)\mathrm{d}\boldsymbol{f}_t \tag{4}$$

where $\mathcal{I}(\boldsymbol{f}_t)$ is the Fisher information matrix:

$$\mathcal{I}(\boldsymbol{f}_t) = \mathbb{E}_p\big[\nabla_{\boldsymbol{f}_t} \log p \nabla_{\boldsymbol{f}_t}^\top \log p\big] . \tag{5}$$

For the Gaussian transition kernel in Eq. (3), the log-likelihood as a function of $\boldsymbol{f}_t$ is given by

$$\log p(\boldsymbol{x}_{t+\mathrm{d}t}|\boldsymbol{x}_t; \boldsymbol{f}_t) \approx \frac{-1}{2g_t^2\mathrm{d}t}\big\|\boldsymbol{x}_{t+\mathrm{d}t} - \boldsymbol{x}_t - \boldsymbol{f}_t\mathrm{d}t\big\|_2^2 . \tag{6}$$

Taking the gradient with respect to $\boldsymbol{f}_t$ yields

$$\nabla_{\boldsymbol{f}_t} \log p = \frac{1}{g_t^2}\big(\boldsymbol{x}_{t+\mathrm{d}t} - \boldsymbol{x}_t - \boldsymbol{f}_t\mathrm{d}t\big) . \tag{7}$$

Under the transition kernel, the residual $\boldsymbol{x}_{t+\mathrm{d}t} - \boldsymbol{x}_t - \boldsymbol{f}_t\mathrm{d}t$ follows a Gaussian distribution $\mathcal{N}(\boldsymbol{0}, g_t^2\mathrm{d}t\,\boldsymbol{I})$.

Therefore, the Fisher information matrix in the drift space is

$$\mathcal{I}(\boldsymbol{f}_t) = \mathbb{E}\big[\nabla_{\boldsymbol{f}_t} \log p \nabla_{\boldsymbol{f}_t}^\top \log p\big] = \frac{\mathrm{d}t}{g_t^2}\boldsymbol{I} . \tag{8}$$

Substituting Eq. (8) into Eq. (4), the Fisher-Rao line element induced by an infinitesimal change $\mathrm{d}\boldsymbol{f}_t$ becomes

$$\mathrm{d}s^2 = \frac{\mathrm{d}t}{g_t^2}\|\mathrm{d}\boldsymbol{f}_t\|_2^2 . \tag{9}$$

Essentially, this Riemannian line element quantifies the statistical displacement on the manifold induced by fluctuations in the drift field. It formalizes the intuition that the transition kernel becomes increasingly sensitive to perturbations in $\boldsymbol{f}_t$ as the noise scale $g_t$ diminishes.

### 3.3. Drift Variation Score

While the Fisher-Rao line element $\mathrm{d}s^2$ in Eq. (9) measures the infinitesimal statistical distance, it is an incremental quantity that vanishes as $\mathrm{d}t \to 0$. For the purpose of adaptive step-size control, we focus on the intensity of information variation per unit time, which is defined by the ratio:

$$V_t = \frac{\mathrm{d}s^2}{\mathrm{d}t} = \frac{\|\mathrm{d}\boldsymbol{f}_t\|_2^2}{g_t^2} . \tag{10}$$

In a discrete numerical solver, let $\{t_k\}_{k=0}^N$ be the adaptive time discrete sequence, where $\boldsymbol{x}_k$ denotes the state at time $t_k$. The infinitesimal change $\mathrm{d}\boldsymbol{f}_t$ is approximated by the finite difference between successive drift evaluations along the sampling trajectory: $\Delta\boldsymbol{f}_{t_k} = \boldsymbol{f}(\boldsymbol{x}_k, t_k) - \boldsymbol{f}(\boldsymbol{x}_{k-1}, t_{k-1})$. Substituting this into Eq. (10) yields the **Drift Variation Score (DVS)**, $V_k$, as a discrete surrogate for the information variation intensity:

$$V_k = \frac{\|\boldsymbol{f}(\boldsymbol{x}_k, t_k) - \boldsymbol{f}(\boldsymbol{x}_{k-1}, t_{k-1})\|_2^2}{g_{t_k}^2} . \tag{11}$$

Unlike the physical time $t$, the Fisher-Rao arc length provides a natural geometric coordinate that reflects the actual rate of distributional change. An ideal numerical scheme should adapt its step size to the local complexity of the dynamics. By maintaining approximately constant increments in the informational distance, the sampler ensures that each integration step contributes a consistent amount of informational progress:

$$\Delta s_k^2 = V_k \cdot \Delta t_k \approx \text{constant} . \tag{12}$$

This condition implies that the step size $\Delta t_k$ should be inversely proportional to the DVS, providing a precise geometric signal for adaptive time-stepping:

*Step-Size Shrinkage (High DVS)*: A large $V$ indicates that the drift field is fluctuating rapidly relative to the noise level. Geometrically, this corresponds to regions of high information-geometric curvature where the sampling trajectory is turning sharply on the manifold. In such cases, the solver must proactively shrink the step size $\Delta t$ to ensure the discrete update tracks the SDE.

*Step-Size Expansion (Low DVS)*: A small $V$ implies that the transition kernels are evolving relatively smoothly, suggesting a flat region on the statistical manifold. In these stable regimes, the solver can safely expand the step size $\Delta t$ to accelerate the sampling process by bypassing redundant evaluations without compromising numerical accuracy.

### 3.4. DVS-driven adaptive sampler for Graph

For graph-structured data $\mathbf{G} = (\mathbf{X}, \mathbf{A})$, the sampling process entails a coupled evolution of node features $\mathbf{X}$ and adjacency structures $\mathbf{A}$. Given the divergent denoising speeds of these two components (Jo et al., 2022), a uniform discretization often fails to capture the intricate local geometry. Following the information-geometric derivation in Section 3.2, we propose a multi-component *Drift Variation Score* (DVS) to monitor the instantaneous velocity of transition distributions on the statistical manifold. At each discrete step $k$, the DVS for nodes and edges is computed as:

$$\begin{cases} V_{\mathbf{X},k} = \dfrac{\|\boldsymbol{f}(\mathbf{X}_k, t_k) - \boldsymbol{f}(\mathbf{X}_{k-1}, t_{k-1})\|_2^2}{g_{t_k}^2} \\[2mm] V_{\mathbf{A},k} = \dfrac{\|\boldsymbol{f}(\mathbf{A}_k, t_k) - \boldsymbol{f}(\mathbf{A}_{k-1}, t_{k-1})\|_2^2}{g_{t_k}^2} \end{cases} \tag{13}$$

To prevent the adaptive signal from being corrupted by the stochastic nature of the diffusion process, we introduce a global variation state, denoted as $\overline{V}$, which serves as a moving memory of the historical trajectory variability. We synchronize the current DVS values with this state using an exponential moving average (EMA):

$$\begin{cases} \overline{V}_{\mathbf{X}} \leftarrow (1-\alpha) \cdot \overline{V}_{\mathbf{X}} + \alpha \cdot V_{\mathbf{X},k} \\[1mm] \overline{V}_{\mathbf{A}} \leftarrow (1-\alpha) \cdot \overline{V}_{\mathbf{A}} + \alpha \cdot V_{\mathbf{A},k} \end{cases} \tag{14}$$

**Algorithm 1** DVS-driven Adaptive Sampler for Graphs.

1: **Input:** Initial states $\mathbf{X}_0, \mathbf{A}_0 \sim p_0$, time $T$, hyperparameters $\kappa_{\text{ref}}, \gamma$, and a solver $\mathcal{S} \in \{\text{Euler, Heun}\}$ .

2: **Initialize:** $t \leftarrow 0, \overline{V} \leftarrow 0$, and $k \leftarrow 1$ .

3: **while** $t < T$ **do**

4:     Compute $V_{\mathbf{X},k}, V_{\mathbf{A},k}$ ;                 ▷ Eq. (13)

5:     Update $\overline{V}_{\mathbf{X}}, \overline{V}_{\mathbf{A}}$ ;                        ▷ Eq. (14)

6:     Compute $\Delta t_{\mathbf{X},k}, \Delta t_{\mathbf{A},k}$ ;                 ▷ Eq. (15)

7:     $\Delta t_k \leftarrow \min(\Delta t_{\mathbf{X},k}, \Delta t_{\mathbf{A},k})$ ;

8:     $\overline{V}_{\mathbf{X}}, \overline{V}_{\mathbf{A}} \leftarrow \gamma \cdot (\overline{V}_{\mathbf{X}} + \overline{V}_{\mathbf{A}})$ ;

9:     $(\mathbf{X}_k, \mathbf{A}_k) \leftarrow \text{SolverStep}(\mathbf{X}_{k-1}, \mathbf{A}_{k-1}, \Delta t_k; \mathcal{S})$ ;

10:     $t \leftarrow t + \Delta t_k, \quad k \leftarrow k + 1$ ;

11: **end while**

12: **Output:** Final graph state $(\mathbf{X}_T, \mathbf{A}_T)$ .

---

where the smoothing coefficient is fixed at $\alpha = 0.2$. This empirical setting ensures a robust transition between time steps, effectively filtering high-frequency noise while remaining responsive to structural shifts in the graph.

Based on the smoothed DVS, the component-wise adaptive step sizes are derived through a power-law scaling rule:

$$\begin{cases} \Delta t_{\mathbf{X},k} = \text{clip}\left(\Delta t_{\text{base}}\left(\frac{\kappa_{\text{ref}}}{\overline{V}_{\mathbf{X}}}\right)^{\beta}, \Delta t_{\min}, \Delta t_{\max}\right) \\ \Delta t_{\mathbf{A},k} = \text{clip}\left(\Delta t_{\text{base}}\left(\frac{\kappa_{\text{ref}}}{\overline{V}_{\mathbf{A}}}\right)^{\beta}, \Delta t_{\min}, \Delta t_{\max}\right) \end{cases} \quad (15)$$

where $\kappa_{\text{ref}}$ acts as the reference curvature and the clipping operator $\text{clip}(x, a, b) = \min(\max(x, a), b)$ bounds the step size within $[\Delta t_{\min}, \Delta t_{\max}]$. The sensitivity exponent $\beta$ is fixed at 0.5, which provides a square-root damping effect on step size fluctuations. This mechanism forces the sampler to contract the step size in high-curvature regions (high DVS) to maintain numerical fidelity, while expanding it in stable regimes (low DVS) to accelerate information flow.

To guarantee global stability across the entire graph, the final advancement step $\Delta t_k$ follows the limiting component (the bottleneck effect), ensuring that neither node nor edge updates exceed the local stability bound: $\Delta t_k = \min(\Delta t_{\mathbf{X},k}, \Delta t_{\mathbf{A},k})$. After the update, the variation state $\overline{V}$ is refreshed via an aggregation factor $\gamma$: $\overline{V}_{\mathbf{X}}, \overline{V}_{\mathbf{A}} \leftarrow \gamma \cdot (\overline{V}_{\mathbf{X}} + \overline{V}_{\mathbf{A}})$. In this framework, $\gamma$ functions as a global feedback gain that governs the coupling strength between node and edge dynamics on the manifold. By modulating $\gamma$, the sampler can balance the contribution of multi-modal signals to the future adaptive schedule.

In our experiments, we employ the DVS sampler summarized in Algorithm 1. This closed-loop control ensures that each step covers an approximately constant information distance on the Riemannian statistical manifold formed by

the family of transition distributions. Crucially, unlike uniform discretization where the total number of steps is a pre-defined hyperparameter, the step count in our framework is an adaptive outcome of the sampling trajectory's intrinsic geometry. By autonomously expanding the time increment $\Delta t$ during stable regimes and contracting it during critical structural transitions, the sampler traverses the time horizon $[0, T]$ more efficiently, often concluding in significantly fewer iterations than a fixed-step baseline while preserving higher numerical fidelity.

## 4. Experiments

### 4.1. Experimental Setup

**Models**  We evaluate the proposed DVS-driven sampling strategy on two representative graph generative models. Our primary experiments are conducted on **GruM** (Jo et al., 2024), a graph diffusion model built upon Ornstein-Uhlenbeck (OU) (Mazzolo, 2017) bridges, which provides a continuous-time formulation for graph-valued stochastic processes. In addition, we perform supplementary experiments on **GDSS** (Jo et al., 2022), a diffusion-based graph generative model with a distinct forward noising and reverse denoising mechanism, to further assess the generality of our approach. For both models, we strictly use the original pretrained checkpoints released by the respective authors, without any modification to the training objective, model architecture, or learned parameters.

**Sampling Methods**  Throughout our experiments, we evaluate the proposed approach across two representative numerical integrators: the first-order **Euler-Maruyama** solver and the second-order **Heun** solver (Hoogeboom et al., 2022). For each integrator, we compare three distinct time-stepping schedules: **Fixed-Step**, **Quadratic** (Song et al., 2021a), and our proposed **DVS** (Drift Variation Score). While Fixed-Step serves as the baseline using uniform discretization, the Quadratic schedule (Song et al., 2021a) is a heuristic power-law strategy that redistributes a fixed total step budget by employing larger, sparser steps in the high-noise regime to allow for denser refinement in the low-noise regime. In contrast, our DVS sampler adaptively determines non-uniform time increments by monitoring the local curvature of the statistical manifold to ensure approximately constant information progress at each step. Consequently, DVS-driven variants, such as DVS-Euler and DVS-Heun, preserve the per-step computational cost of their fixed-step counterparts while achieving a more balanced distributional evolution under a comparable number of function evaluations (NFE).

**Implementation Details**  For computational efficiency and stability considerations, we apply DVS sampler only over selected portions of the sampling trajectory on certain datasets (See Appendix B.1). Specifically, in these

*Table 1.* **Generation results on 2D molecule datasets using the GRUM model.** Best results are highlighted in **bold**. The DVS rows (Ours) consistently show superior fidelity across distributional (FCD) metric.

| Solver | Method | QM9 | | | ZINC250k | | |
|--------|--------|-----------|-------|---------|-----------|-------|---------|
| | | Valid (%) ↑ | FCD ↓ | NSPDK ↓ | Valid (%) ↑ | FCD ↓ | NSPDK ↓ |
| Euler-Maruyama | Fixed-Step (Jo et al., 2024) | 99.43 | 0.107 | **0.0002** | 98.34 | 2.207 | 0.0016 |
| | Quadratic (Song et al., 2021a) | 99.46 | 0.107 | **0.0002** | 98.33 | 2.194 | 0.0016 |
| | **DVS (Ours)** | **99.53** | **0.095** | **0.0002** | **98.51** | **2.092** | **0.0015** |
| Heun | Fixed-Step (Jo et al., 2024) | 99.47 | 0.109 | **0.0002** | 98.39 | 2.182 | **0.0015** |
| | Quadratic (Song et al., 2021a) | 99.44 | 0.107 | **0.0002** | **98.47** | 2.154 | **0.0015** |
| | **DVS (Ours)** | **99.55** | **0.099** | **0.0002** | 98.45 | **2.086** | **0.0015** |

*Table 2.* **Generation results on the QM9 dataset using the GDSS model.** Our DVS sampler achieves consistent improvements across all metrics. Best results are in **bold**.

| Method | Valid (%) ↑ | FCD ↓ | NSPDK ↓ |
|--------|-----------|-------|---------|
| Euler-Maruyama | 94.96 | 2.551 | 0.0036 |
| **DVS (Ours)** | **95.11** | **2.482** | **0.0035** |

settings, the adaptive strategy is activated within predefined time intervals where the sampling dynamics exhibit rapid variation, while a fixed step size is used elsewhere. The choice of adaptive intervals is determined empirically and kept fixed across all compared methods to ensure fairness. We describe further implementation details including the hyperparameters in Appendix B.1.

### 4.2. Molecule Generation

**Datasets and Metrics** We evaluate our DVS sampler on two widely used molecular graph generation benchmarks: QM9 (Ramakrishnan et al., 2014) and ZINC250k (Irwin et al., 2012). QM9 consists of small organic molecules with up to nine heavy atoms, serving as a standard testbed for controlled molecular generation, while ZINC250k contains more diverse and structurally complex drug-like molecules, posing greater challenges in both validity and distributional fidelity. We assess generation quality using three complementary metrics. Specifically, consistent with the evaluation benchmarks of existing graph diffusion models, we report the Validity without correction (Valid) (Shi et al., 2020; Luo et al., 2021) to evaluate fundamental chemical reliability. To measure the distributional alignment between the generated and real molecular distributions, we employ the Fréchet ChemNet Distance (FCD) (Preuer et al., 2018). Furthermore, we include the NSPDK (Costa & De Grave, 2010) metric to quantify the fine-grained structural similarity of local graph patterns between the generated samples and the reference data. Together, these metrics provide a comprehensive evaluation of the impact of different strategies.

**Results** The generation performance on molecular benchmarks is summarized in *Tables* 1 and 2. Our DVS sampler consistently achieves superior fidelity across both datasets

compared to all baseline schedules. In both Euler and Heun settings, the DVS approach outperforms the Fixed-Step and Quadratic schedules. While the Quadratic schedule offers slight gains over the linear baseline by manually densifying steps near the data distribution, it remains a blind heuristic that fails to account for the specific dynamical stiffness of different molecular structures. In contrast, DVS dynamically senses the information variation intensity, allowing for a more precise resolution of the crystallization phase where discrete chemical bonds are formed. A significant observation is that DVS-Euler often rivals or even surpasses the performance of Fixed-Step Heun across multiple metrics. This underscores a core insight of our work: for graph-structured data, the strategic allocation of steps along the statistical manifold to achieve equal arc-length progression is more impactful than the local error-order of the numerical integrator. Further visualizations of the generated molecules are provided in Appendix D.1.

**Convergence Analysis** *Figure* 2 presents a detailed comparison of the FCD convergence behavior between the original fixed-step Euler-Maruyama sampler and DVS-driven Euler-Maruyama sampler on the QM9 dataset, using the same GRUM-based graph diffusion model. Euler employs a uniform discretization of the reverse-time SDE with a fixed budget of 1000 sampling steps. In contrast, our method dynamically adjusts the integration step size based on local reverse-time dynamics, resulting in a total of **787** sampling steps. Despite using fewer steps, DVS sampler converges faster and reaches a lower final FCD value of **0.0948**, compared to 0.1065 achieved by Euler. The simultaneous reduction in steps and error confirms that DVS effectively reallocates the computational budget toward sensitive structural transitions. As shown in the curves, Euler exhibits a relatively slow decay in the early and intermediate stages of sampling, suggesting that a large fraction of uniform steps contribute marginally to distributional refinement. In contrast, our method demonstrates a steeper decline during high-variation phases and a smoother, more stable convergence in the late-stage regime, as highlighted in the inset. These results indicate that DVS sampler can improve both sampling efficiency and final generation quality.

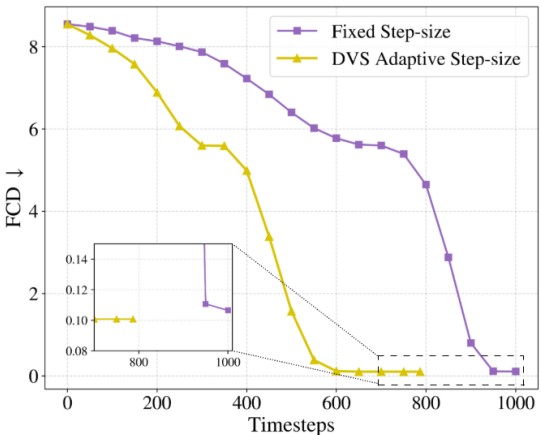

*Figure 2.* **Convergence behavior of FCD during sampling on QM9 using GRUM.** By dynamically sensing the drift variation, our method converges 21.3% faster than the fixed-step baseline while attaining a higher distributional fidelity (lower FCD). The inset confirms that the DVS-driven trajectory maintains smoother convergence during the final phase of molecule generation.

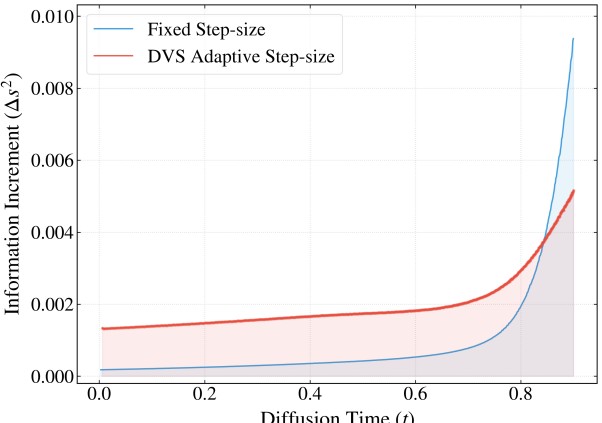

*Figure 3.* **Evolution of the information-geometric increment ($\Delta s^2$) on QM9 using GRUM.** We compare the information progress of the fixed-step Euler-Maruyama sampler and the DVS-driven Euler-Maruyama sampler. For the fixed-step sampler, the information increment remains small at early stages and increases sharply near the end, whereas the adaptive sampler maintains an approximately *equal arc-length* progression throughout sampling.

**Arc-Length Analysis** In an ideal geometric integrator, the sampler should advance with a constant informational increment $\Delta s^2 \approx$ const, ensuring each discrete step covers an equal distance in terms of distributional change. *Figure* 3 visually verifies this principle. The fixed step-size approach (blue curve) exhibits a highly imbalanced distribution of informational progress: in the early and intermediate stages, the information increment remains disproportionately small, indicating that uniform time steps yield negligible statistical progress during these phases. However, this is followed by an exponential explosion in the final stage, where the sampler is forced to rush through regions of extreme curvature within a single step, inevitably introducing significant numerical errors. In sharp contrast, the DVS-driven trajectory (red curve) maintains a remarkably consistent and near-uniform informational progress throughout the sampling process, thereby allowing that each step contributes meaningfully to the generation. By dynamically adjusting the step size $\Delta t$ to compensate for the varying manifold curvature, the DVS controller ensures that the information increment is neither too small to be inefficient in the early stages, nor too large to be unstable in the late stages. While DVS is significantly more stable, a subtle upward trend appears at the end due to the numerical lower bound $\Delta t_{\min}$. As curvature surges during structural crystallization, the sampler hits this minimum threshold and can no longer fully offset the drift variation. Nevertheless, DVS successfully flattens the informational profile, suppressing the exponential explosion seen in the baseline. The plot is truncated slightly before the absolute endpoint $t = T$. Near this limit, the informational increment of the fixed-step baseline becomes so divergent that its inclusion would compromise the visual resolution of the comparison.

### 4.3. General Graph Generation

**Datasets and Metrics** We evaluate the DVS sampler on three standard benchmarks for general graph generation: Planar (Kipf & Welling, 2017), SBM (Holland et al., 1983), and Ego-small (Sen et al., 2008). Planar and SBM are synthetic graph datasets designed to test global and community-level structural modeling, while Ego-small is a real-world ego-centric social graph dataset with diverse local neighborhoods. To assess the quality of generated graphs, we adopt a set of widely used distribution-level metrics that capture both local and global structural properties. Specifically, we evaluate the Maximum Mean Discrepancy (MMD) (Costa & De Grave, 2010) between the generated and reference graphs with respect to four complementary topological properties: degree (Deg.), clustering coefficient (Clus.), count of orbits with 4 nodes (Orbit), and the eigenvalues of the graph Laplacian (Spec.) (Martinkus et al., 2022).

**Results** The evaluation results on general graph datasets are summarized in *Tables* 3 and 4. Our DVS sampler consistently outperforms both fixed-step and heuristic quadratic schedules across the vast majority of structural metrics. While the Quadratic schedule provides a better distribution of steps than the linear baseline in certain scenarios, it still lags behind the DVS-based approach. Our method achieves significantly lower Spectral and Orbit MMD values, implying that dynamic feedback is more effective at capturing the global topology and local subgraph motifs than static power-law prescriptions. This demonstrates that DVS can uniquely sense and resolve the instantaneous stiffness of the reverse-time SDE. On the Ego-small dataset, DVS-driven sampling shows a clear advantage in reducing

*Table 3.* **Generation results on the general graph datasets using the GruM model.** The best results are highlighted in **bold**. Our DVS sampler consistently achieves lower MMD values across most structural properties, demonstrating its superior ability to preserve complex topological characteristics compared to non-adaptive baselines.

| Solver | Method | Planar | | | | SBM | | | |
|---|---|---|---|---|---|---|---|---|---|
| | | Deg. ↓ | Clus. ↓ | Orbit ↓ | Spec. ↓ | Deg. ↓ | Clus. ↓ | Orbit ↓ | Spec. ↓ |
| Euler-Maruyama | Fixed-Step (Jo et al., 2024) | 0.0002 | 0.0300 | 0.0013 | 0.0060 | 0.0006 | 0.0498 | 0.0455 | 0.0051 |
| | Quadratic (Song et al., 2021a) | 0.0006 | 0.0352 | 0.0046 | 0.0068 | **0.0005** | 0.0487 | 0.0386 | 0.0038 |
| | **DVS (Ours)** | **0.0001** | **0.0283** | **0.0010** | **0.0052** | 0.0007 | **0.0477** | **0.0382** | **0.0030** |
| Heun | Fixed-Step (Jo et al., 2024) | **0.0000** | 0.0307 | 0.0012 | 0.0059 | 0.0008 | 0.0496 | 0.0445 | 0.0046 |
| | Quadratic (Song et al., 2021a) | 0.0003 | 0.0289 | 0.0045 | 0.0068 | **0.0003** | 0.0492 | 0.0405 | 0.0035 |
| | **DVS (Ours)** | **0.0000** | **0.0268** | 0.0011 | **0.0049** | 0.0007 | **0.0482** | **0.0375** | **0.0028** |

*Table 4.* **Generation results on the Ego-small dataset using the GDSS model.** Best results are highlighted in **bold**.

| Method | Deg. ↓ | Clus. ↓ | Orbit ↓ | Spec. ↓ |
|---|---|---|---|---|
| Euler-Maruyama | 0.0198 | 0.0184 | **0.0130** | 0.0369 |
| **DVS (Ours)** | **0.0195** | **0.0149** | **0.0130** | **0.0331** |

*Table 5.* Sensitivity analysis of the scaling factor $\gamma$ on the QM9 dataset using the GruM model.

| $\gamma$ | NFE | Valid ↑ | FCD ↓ | Scaf. ↑ | Frag. ↑ |
|---|---|---|---|---|---|
| Euler | 1000 | 0.9943 | 0.1065 | 0.9341 | 0.9842 |
| 0.10 | 706 | 0.9937 | 0.1050 | 0.9370 | 0.9842 |
| 0.15 | 724 | 0.9945 | 0.1037 | 0.9409 | 0.9843 |
| 0.20 | 745 | 0.9947 | **0.0976** | 0.9415 | 0.9818 |
| 0.25 | 770 | 0.9956 | 0.1028 | **0.9455** | **0.9844** |
| 0.30 | 799 | **0.9957** | 0.1065 | 0.9435 | 0.9843 |
| 0.35 | 836 | 0.9951 | 0.1043 | 0.9428 | 0.9839 |

Cluster and Spectral MMD. This improvement indicates that by adapting to the local geometry of the distribution space, our sampler better preserves the dense connectivity and community structures characteristic of real-world social networks. The consistency of these results across both synthetic and real datasets highlights the robustness of the information-geometric criterion. We further visualize the structural evolution of the graphs in Appendix D.2.

### 4.4. Ablation Studies

**Aggregation scaling factor $\gamma$** In this section, we investigate the sensitivity of the aggregation scaling factor $\gamma$, which modulates the feedback intensity of the DVS controller by scaling the global variation state $\overline{V}$ to adaptively adjust the integration step size. As summarized in *Table 5*, our experiments on the QM9 dataset using the GRUM model reveal a clear positive correlation between $\gamma$ and the total number of sampling steps, where increasing $\gamma$ from 0.10 to 0.35 results in a progression from 706 to 836 timesteps. This behavior aligns with the theoretical expectation that a larger $\gamma$ induces a more conservative feedback, forcing the sampler to resolve local dynamics with finer discretization. Crucially, across the entire tested range, the DVS sampler consumes significantly fewer function evaluations (NFE) than the fixed-step Euler baseline of 1000 steps, while simultaneously achieving superior generation quality. We observe that the Fréchet ChemNet Distance (FCD) reaches its optimum of 0.0976 at $\gamma = 0.20$, representing a substantial improvement over the baseline's 0.1065 despite a 25.5% reduction in computational cost. Furthermore, structural metrics such as Scaffold (Scaf.) and Fragment (Frag.) similarities peak at $\gamma = 0.25$, demonstrating that our method effectively preserves chemically meaningful patterns.

## 5. Conclusion

In this paper, we proposed the DVS-driven adaptive sampler to address the inherent challenges of non-uniform dynamics and the desynchronization between node and edge denoising processes in graph diffusion models. Our approach is built upon a rigorous information-geometric framework that treats the evolution of transition kernels induced by numerical integration as a curve on a Riemannian statistical manifold. By utilizing the Fisher-Rao metric to derive the Drift Variation Score (DVS), the sampler can monitor the instantaneous intensity of information variation along the sampling trajectory. This feedback mechanism allows the integrator to automatically contract step sizes during rapidly varying stages and expand them in stable regions, effectively maintaining approximately constant informational progress per step. Comprehensive experiments on molecular and general graph benchmarks demonstrate that the DVS sampler consistently improves generation quality and structural validity compared to standard fixed-step schedules and pre-defined non-uniform strategies, such as the quadratic schedule. As a training-free and plug-and-play module, our method can be seamlessly integrated into existing diffusion frameworks to provide robust and efficient sampling without additional model retraining. For future work, we aim to extend this information-geometric criterion to domains such as 3D geometric modeling, where disparate data components evolve at vastly different rates and require the precise, adaptive control offered by our framework.

## Impact Statement

This paper presents work whose goal is to advance the field of Machine Learning. There are many potential societal consequences of our work, none which we feel must be specifically highlighted here.

## Acknowledgments

This work was supported by the National Natural Science Foundation of China under Grant No. 62176108.

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

# Appendix

**Organization**  The appendices are organized as follows: In **Section A**, we provide the **detailed mathematical derivations** for the information-geometric framework, including the derivation of the joint Fisher Information Matrix for location-scale transition kernels and the numerical justification for focusing on drift variation. In **Section B**, we elaborate on the **implementation details**, specifying the common and dataset-specific hyperparameter configurations along with the detailed pseudocode for the DVS-driven Euler and Heun samplers. In **Section C**, we present **additional experimental results**, including statistical significance analysis and a detailed runtime efficiency analysis. In **Section D**, we provide **qualitative visualizations** of the final generated molecules and the structural evolution snapshots of graph topology throughout the sampling process.

## A. Detailed Mathematical Derivations

### A.1. Fisher Information Matrix for the Location-Scale Transition Kernel

In the main text, we primarily focused on the information variation induced by the drift field $\boldsymbol{f}$. To provide a more rigorous and complete geometric foundation, we here derive the joint Fisher Information Matrix (FIM) by considering both the drift vector $\boldsymbol{f} \in \mathbb{R}^D$ and the diffusion coefficient $g \in \mathbb{R}^+$ as local parameters of the transition kernel.

Let $\theta = [\boldsymbol{f}^\top, g]^\top$ be the $(D+1)$-dimensional joint parameter vector. The Gaussian transition density $p(\boldsymbol{x}_{t+\mathrm{d}t} \mid \boldsymbol{x}_t; \theta)$ induced by the numerical solver for a small time increment $\mathrm{d}t$ is expressed as:

$$p(\boldsymbol{x}_{t+\mathrm{d}t} \mid \boldsymbol{x}_t; \boldsymbol{f}, g) = \frac{1}{(2\pi g^2 \mathrm{d}t)^{D/2}} \exp\left(-\frac{\|\boldsymbol{x}_{t+\mathrm{d}t} - \boldsymbol{x}_t - \boldsymbol{f}\mathrm{d}t\|_2^2}{2g^2 \mathrm{d}t}\right). \tag{16}$$

Defining the residual vector as $\boldsymbol{r} = \boldsymbol{x}_{t+\mathrm{d}t} - \boldsymbol{x}_t - \boldsymbol{f}\mathrm{d}t$, the log-likelihood function $\mathcal{L}(\theta) = \log p$ is given by:

$$\mathcal{L}(\theta) = -\frac{D}{2}\log(2\pi \mathrm{d}t) - D\log g - \frac{\boldsymbol{r}^\top \boldsymbol{r}}{2g^2 \mathrm{d}t}. \tag{17}$$

**Score Functions (First-Order Derivatives)**  The score functions with respect to the parameters $\boldsymbol{f}$ and $g$ represent the sensitivity of the log-likelihood:

$$\nabla_{\boldsymbol{f}}\mathcal{L} = \frac{\partial \mathcal{L}}{\partial \boldsymbol{r}}\frac{\partial \boldsymbol{r}}{\partial \boldsymbol{f}} = \left(-\frac{\boldsymbol{r}}{g^2 \mathrm{d}t}\right) \cdot (-\mathrm{d}t) = \frac{\boldsymbol{r}}{g^2}, \tag{18}$$

$$\nabla_g \mathcal{L} = -\frac{D}{g} + \frac{\boldsymbol{r}^\top \boldsymbol{r}}{g^3 \mathrm{d}t}. \tag{19}$$

**Structure of the Joint Fisher Information Matrix**  Let $\nabla_\theta \mathcal{L} = [(\nabla_{\boldsymbol{f}}\mathcal{L})^\top, \nabla_g \mathcal{L}]^\top$ be the joint score vector. The FIM $\mathcal{I}(\theta)$ is defined as the expected outer product of these components:

$$\mathcal{I}(\boldsymbol{f}, g) = \mathbb{E}\left[(\nabla_\theta \mathcal{L})(\nabla_\theta \mathcal{L})^\top\right] = \begin{bmatrix} \mathcal{I}_{ff} & \mathcal{I}_{fg} \\ \mathcal{I}_{gf} & \mathcal{I}_{gg} \end{bmatrix}. \tag{20}$$

*(i) Drift Component $\mathcal{I}_{ff}$:* Since the residual follows a Gaussian distribution $\boldsymbol{r} \sim \mathcal{N}(\boldsymbol{0}, g^2 \mathrm{d}t \mathbf{I})$, its second moment is $\mathbb{E}[\boldsymbol{r}\boldsymbol{r}^\top] = g^2 \mathrm{d}t \mathbf{I}$. Substituting Eq. (18) into the FIM definition:

$$\mathcal{I}_{ff} = \mathbb{E}[(\nabla_{\boldsymbol{f}}\mathcal{L})(\nabla_{\boldsymbol{f}}\mathcal{L})^\top] = \frac{1}{g^4}\mathbb{E}[\boldsymbol{r}\boldsymbol{r}^\top] = \frac{\mathrm{d}t}{g^2}\mathbf{I}. \tag{21}$$

*(ii) Information Orthogonality $\mathcal{I}_{fg} = \boldsymbol{0}$:* The off-diagonal block represents the information coupling between drift and diffusion variation:

$$\mathcal{I}_{fg} = \mathbb{E}[(\nabla_{\boldsymbol{f}}\mathcal{L})(\nabla_g \mathcal{L})] = \mathbb{E}\left[\frac{\boldsymbol{r}}{g^2}\left(\frac{\boldsymbol{r}^\top \boldsymbol{r}}{g^3 \mathrm{d}t} - \frac{D}{g}\right)\right]. \tag{22}$$

Because the transition kernel is centrally symmetric, the expectations of the odd-order moments of $r$ (specifically the 1st and 3rd order) are identically zero. Thus, $\mathcal{I}_{fg} = 0$, meaning that the drift (location) and noise scale are *information-theoretically orthogonal*.

*(iii) Diffusion Component $\mathcal{I}_{gg}$:* We compute this scalar component using the expected second-order derivative:

$$\mathcal{I}_{gg} = -\mathbb{E}\left[\nabla_g^2 \mathcal{L}\right] = -\mathbb{E}\left[\frac{D}{g^2} - \frac{3r^\top r}{g^4 dt}\right] = \frac{2D}{g^2}. \tag{23}$$

**Riemannian Line Element**  By assembling the block-diagonal components, the squared infinitesimal distance $ds^2$ on the manifold $\mathcal{M}$ is given by the quadratic form:

$$ds^2 = d\theta^\top \mathcal{I}(\theta) d\theta = \underbrace{\frac{dt}{g^2}\|d\boldsymbol{f}\|_2^2}_{\text{Drift Contribution}} + \underbrace{\frac{2D}{g^2}(dg)^2}_{\text{Noise Contribution}}. \tag{24}$$

## A.2. Rationale for Neglecting $g$ in DVS

While the Riemannian line element $ds^2$ theoretically incorporates contributions from both the drift field $\boldsymbol{f}$ and the diffusion coefficient $g_t$, we demonstrate that the noise-related term is asymptotically negligible and provides limited information for adaptive sampling. This design choice is justified through scaling analysis and empirical evidence.

**SDE-Based Scaling Decomposition.**  We consider the reverse-time SDE in Eq. (1). The drift field $\boldsymbol{f}$ is a learned response of the neural network that adapts to the instantaneous state $\boldsymbol{x}$. Applying Itô's lemma, its infinitesimal variation can be decomposed as:

$$d\boldsymbol{f} = \left(\frac{\partial \boldsymbol{f}}{\partial t} + \mathbf{J}_{\boldsymbol{f}}\boldsymbol{f} + \frac{1}{2}g^2 \Delta_{\boldsymbol{x}}\boldsymbol{f}\right) dt + g\mathbf{J}_{\boldsymbol{f}} d\bar{\boldsymbol{w}}, \tag{25}$$

where $\mathbf{J}_{\boldsymbol{f}} = \nabla_{\boldsymbol{x}}\boldsymbol{f}$ is the Jacobian of the drift. Taking the expectation of the squared norm, the leading-order term is:

$$\mathbb{E}\|d\boldsymbol{f}\|_2^2 = g^2 \mathrm{Tr}(\mathbf{J}_{\boldsymbol{f}}^\top \mathbf{J}_{\boldsymbol{f}}) dt + \mathcal{O}(dt^2), \tag{26}$$

which implies the magnitude of drift variation scales as $\|d\boldsymbol{f}\|_2 = \mathcal{O}(\sqrt{dt})$.

In contrast, the diffusion coefficient $g$ is typically a fixed, predefined noise schedule (e.g., linear or cosine), encoding only time-dependent structure. Its variation is governed by its first-order derivative:

$$dg = \dot{g}dt \implies |dg| = \mathcal{O}(dt). \tag{27}$$

Comparing the two scales, the ratio of drift variation to noise variation yields:

$$\frac{\|d\boldsymbol{f}\|_2}{|dg|} \approx \frac{g\sqrt{\mathrm{Tr}(\mathbf{J}_{\boldsymbol{f}}^\top \mathbf{J}_{\boldsymbol{f}})}\sqrt{dt}}{|\dot{g}|dt} = \frac{g\sqrt{\mathrm{Tr}(\mathbf{J}_{\boldsymbol{f}}^\top \mathbf{J}_{\boldsymbol{f}})}}{|\dot{g}|} \cdot \frac{1}{\sqrt{dt}} = \mathcal{O}\left(\frac{1}{\sqrt{dt}}\right) \to \infty \quad \text{as} \quad dt \to 0. \tag{28}$$

This suggests that at an infinitesimal resolution, the information variation is overwhelmingly dominated by the dynamics of the drift field rather than the evolution of the diffusion coefficient.

**Empirical Validation.**  We empirically evaluate the magnitudes of the drift component $V_f = \frac{dt}{g^2}\|d\boldsymbol{f}\|_2^2$ and the noise component $V_g = \frac{2D}{g^2}(dg)^2$ (as defined in Appendix A) throughout the sampling process on the QM9 dataset. As illustrated in *Figure* 4, the information variation intensity induced by the drift field dominates the total Riemannian line element by an overwhelming margin across the entire trajectory. Specifically, the drift variation $V_f$ is consistently several orders of magnitude higher than the noise variation $V_g$, with the mean ratio reaching approximately $3.3 \times 10^6$. This disparity becomes even more pronounced during the structural crystallization phase ($t > 0.85$), where $V_f$ exhibits an exponential surge while $V_g$ remains relatively smooth and low-magnitude. These results demonstrate that the deterministic noise schedule provides limited signal to the adaptive feedback loop, and omitting it ensures that the Drift Variation Score remains a lightweight and robust criterion.

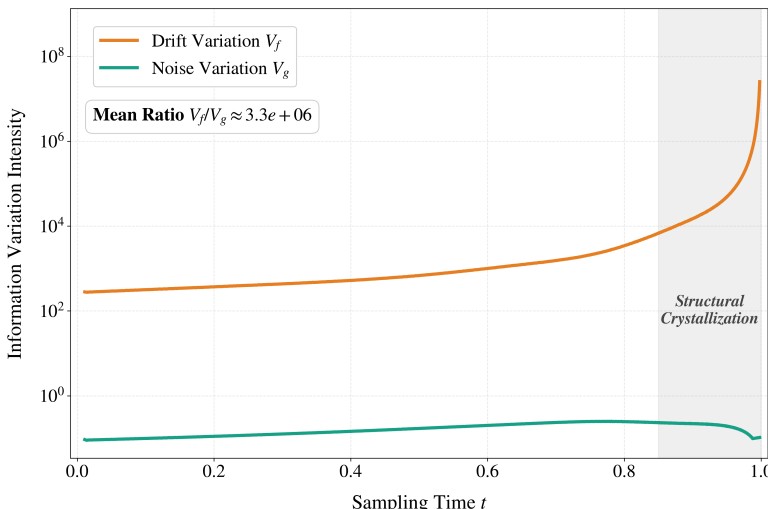

*Figure 4.* **Numerical scale analysis of information variation components on QM9 using the GruM model.** The drift component $V_f$ (red solid line) represents the data-dependent variation of the transition distribution, which dominates the noise component $V_g$ (blue dashed line) by over six orders of magnitude on average. The explosive growth of $V_f$ near $t = 1$ highlights the intense structural transitions (crystallization) that necessitate adaptive step-size control.

*Table 6.* Common hyperparameters and numerical constants for the DVS-driven adaptive sampler.

| Parameter | Symbol | Value | Description |
|---|---|---|---|
| EMA Smoothing | $\alpha$ | 0.2 | Smoothing coefficient for DVS variability. |
| Sensitivity Exponent | $\beta$ | 0.5 | Controls the response to manifold curvature. |
| Base Timestep | $\Delta t_{\text{base}}$ | $1 \times 10^{-3}$ | Initial step size (corresponds to $N = 1000$). |
| Minimum Timestep | $\Delta t_{\text{min}}$ | $2 \times 10^{-4}$ | Lower bound to prevent numerical stalling. |
| Maximum Timestep | $\Delta t_{\text{max}}$ | $5 \times 10^{-3}$ | Upper bound to maintain stability. |
| Stability Constant | $\epsilon_{\text{num}}$ | $1 \times 10^{-12}$ | Prevents division-by-zero in computations. |
| Boundary Tolerance | $\epsilon_{\text{bound}}$ | $1 \times 10^{-6}$ | Ensures robust termination at boundary $T$. |

# B. Implementation Details

This appendix provides comprehensive details regarding the implementation of the DVS-driven adaptive sampler, including the hyperparameter search space, numerical safeguards, and the rationale behind dataset-specific configurations.

## B.1. Hyperparameter Settings for DVS-driven Sampler

To ensure the reproducibility of our results, we provide the detailed hyperparameter configurations used across all benchmarks. The DVS sampler is designed as a modular, training-free component that can be integrated into existing reverse-time SDE solvers with minimal tuning.

**Common Hyperparameters.** *Table* 6 lists the hyperparameters that were kept constant across all datasets and models. We found that the sampler is relatively robust to these settings, and we used the same values for both molecular and general graph generation tasks.

**Dataset-Specific Hyperparameters.** The optimal configurations for the DVS-driven sampler vary across datasets and model architectures due to their distinct dynamical properties. *Table* 7 summarizes the reference curvature $\kappa_{\text{ref}}$, the aggregation factor $\gamma$ for both Euler and Heun solvers, and the specific time intervals where the adaptive strategy is activated.

**Interval Selection Logic.** The active range is an optional design choice instead of an essential component of DVS. This parameter serves as a tunable hyperparameter that governs the trade-off between computational efficiency and performance. It can be optimized on a validation set. Introducing such a range does not alter the underlying geometry-driven mechanism of DVS. The method remains fundamentally general and prior-free, while full-trajectory adaptation serves as the default formulation. The results presented in *Table* 8 provide empirical evidence corroborating this claim. DVS can be directly

*Table 7.* Dataset-specific hyperparameters and adaptive intervals for GruM and GDSS.

| Model | Dataset | $\kappa_{\mathrm{ref}}$ | Euler $\gamma$ | Heun $\gamma$ | Active Range $(t)$ |
|---|---|---|---|---|---|
| GruM | QM9 | 1.0 | 0.22 | 0.23 | $[0, 1]$ |
| | ZINC250k | 5.0 | 0.02 | 0.04 | $[0, 1]$ |
| | Planar | 10.0 | 0.31 | 0.30 | $[0.5, 1.0]$ |
| | SBM | 10.0 | 0.26 | 0.26 | $[0.4, 1.0]$ |
| GDSS | QM9 | 1.0 | 0.68 | — | $[0, 0.2] \cup [0.95, 1]$ |
| | Ego-small | 0.2 | 0.02 | — | $[0.95, 1]$ |

applied over the entire sampling trajectory without manual restriction. On the Planar and SBM datasets using the GruM model, full-trajectory DVS consistently outperforms the fixed-step Euler-Maruyama baseline across all metrics. This demonstrates inherent stability and effectiveness even in the absence of specific range selection.

For molecular datasets such as QM9 and ZINC250k under the GruM framework, the adaptive sampler is applied throughout the entire trajectory from 0 to 1 to ensure maximum fidelity. For synthetic graphs such as Planar and SBM, the DVS controller can be activated during the late stages where structural crystallization occurs and the drift variation is most pronounced. In the GDSS model, the controller targets specific regions of high informational intensity such as the initial noise perturbation phase and the final structural refinement phase. This strategy maximizes efficiency while preserving generation quality.

*Table 8.* **Comparison of Full-Trajectory DVS and Active-Range DVS.** Full-trajectory DVS already outperforms the fixed-step baseline, demonstrating the robustness of the information-geometric criterion without manual range restriction.

| Dataset | Method | Deg. ↓ | Clus. ↓ | Orbit ↓ | Spec. ↓ |
|---|---|---|---|---|---|
| **Planar** | Fixed-Step | 0.0002 | 0.0300 | 0.0013 | 0.0060 |
| | Full DVS (Ours) | 0.0001 | 0.0294 | 0.0013 | 0.0058 |
| | Active-Range DVS (Ours) | 0.0001 | 0.0283 | 0.0010 | 0.0052 |
| **SBM** | Fixed-Step | 0.0006 | 0.0498 | 0.0455 | 0.0051 |
| | Full DVS (Ours) | 0.0002 | 0.0481 | 0.0386 | 0.0049 |
| | Active-Range DVS (Ours) | 0.0007 | 0.0477 | 0.0382 | 0.0030 |

### B.2. Pseudocode for DVS-driven Adaptive Sampler

In this section, we provide the detailed algorithmic implementations for the **DVS-Euler-Maruyama** and **DVS-Heun** samplers, as shown in Algorithms 2 and 3, respectively. These pseudocodes illustrate how the Drift Variation Score (DVS) is computed using cached drift values to maintain high efficiency.

## C. Additional Experimental Results

### C.1. Statistical Significance Analysis

We evaluate statistical significance on the QM9 dataset using the GDSS model. We compare the fixed-step Euler-Maruyama sampler with the DVS-driven Euler-Maruyama sampler across 10 independent runs using identical random seeds. We report mean and standard deviation for each metric, perform paired $t$-tests based on seed-wise results, and compute 95% confidence intervals (CI) for the performance differences. The results in *Table 9* show that all metrics exhibit statistically significant improvements under paired $t$-tests. In particular, FCD shows a substantial and highly significant improvement ($p < 0.001$), indicating consistent gains in distributional fidelity.

### C.2. Efficiency and Runtime Analysis

DVS is a lightweight mechanism that introduces no additional neural network evaluations. The dominant cost of diffusion sampling lies in evaluating the drift network $f_t(x)$, resulting in a total complexity of $\mathcal{O}(K \cdot C_f)$, where $K$ is the number of steps and $C_f$ is the cost of a single forward pass. For graph diffusion models, this cost is typically dominated by dense

---

**Algorithm 2** DVS-Euler-Maruyama Adaptive Sampler for Graphs.

---

1: **Input:** Initial states $\mathbf{X}_0, \mathbf{A}_0 \sim p_0$, time $t, T$, denoising networks $f^{\psi_X}, f^{\psi_A}$, and hyperparameters $\kappa_{\text{ref}}, \gamma$.
2: **Initialize:** $t \leftarrow 0, \overline{V}_{\mathbf{X}}, \overline{V}_{\mathbf{A}} \leftarrow 0$, and step index $k \leftarrow 1$.
3: **while** $t < T - \epsilon_{\text{bound}}$ **do**
4:     $\boldsymbol{f}_{\mathbf{X},k}, \boldsymbol{f}_{\mathbf{A},k} \leftarrow f^{\psi_X}(\mathbf{X}_{k-1}, \mathbf{A}_{k-1}, t), \; f^{\psi_A}(\mathbf{X}_{k-1}, \mathbf{A}_{k-1}, t)$;
5:     **if** $k > 1$ **then**
6:         Compute $V_{\mathbf{X},k}, V_{\mathbf{A},k}$ using $(\boldsymbol{f}_{\mathbf{X},k}, \boldsymbol{f}_{\mathbf{X},k-1})$ and $(\boldsymbol{f}_{\mathbf{A},k}, \boldsymbol{f}_{\mathbf{A},k-1})$;              $\triangleright$ Eq. (13)
7:         Update $\overline{V}_{\mathbf{X}}, \overline{V}_{\mathbf{A}}$ by $V_{\mathbf{X},k}, V_{\mathbf{A},k}$;                         $\triangleright$ Eq. (14)
8:         Compute $\Delta t_{\mathbf{X},k}, \Delta t_{\mathbf{A},k}$ and set $\Delta t_k \leftarrow \min(\Delta t_{\mathbf{X},k}, \Delta t_{\mathbf{A},k})$;          $\triangleright$ Eq. (15)
9:         $\overline{V}_{\mathbf{X},k}, \overline{V}_{\mathbf{A},k} \leftarrow \gamma \cdot (\overline{V}_{\mathbf{X},k} + \overline{V}_{\mathbf{A},k})$;                    $\triangleright$ Update variation state
10:     **else**
11:         $\Delta t_k \leftarrow \Delta t_{\text{base}}$;
12:     **end if**
13:     $\Delta t_k \leftarrow \min(\Delta t_k, T - t)$;
14:     $\mathbf{Z}_{\mathbf{X}}, \mathbf{Z}_{\mathbf{A}} \sim \mathcal{N}(\mathbf{0}, \mathbf{I})$;
15:     $\mathbf{X}_k \leftarrow \mathbf{X}_{k-1} + \boldsymbol{f}_{\mathbf{X},k}\Delta t_k + g_k\sqrt{\Delta t_k}\mathbf{Z}_{\mathbf{X}}$;
16:     $\mathbf{A}_k \leftarrow \mathbf{A}_{k-1} + \boldsymbol{f}_{\mathbf{A},k}\Delta t_k + g_k\sqrt{\Delta t_k}\mathbf{Z}_{\mathbf{A}}$;
17:     $t \leftarrow t + \Delta t_k, \quad k \leftarrow k + 1$;
18: **end while**
19: **Output:** A graph $(\mathbf{X}_T, \mathbf{A}_T)$.

---

**Algorithm 3** DVS-Heun Adaptive Sampler for Graphs.

---

1: **Input:** Initial states $\mathbf{X}_0, \mathbf{A}_0 \sim p_0$, time $t, T$, denoising networks $f^{\psi_X}, f^{\psi_A}$, and hyperparameters $\kappa_{\text{ref}}, \gamma$.
2: **Initialize:** $t \leftarrow 0, \overline{V}_{\mathbf{X}}, \overline{V}_{\mathbf{A}} \leftarrow 0$, and step index $k \leftarrow 1$.
3: **while** $t < T - \epsilon_{\text{bound}}$ **do**
4:     $\boldsymbol{f}_{\mathbf{X},k}^{(1)}, \boldsymbol{f}_{\mathbf{A},k}^{(1)} \leftarrow f^{\psi_X}(\mathbf{X}_{k-1}, \mathbf{A}_{k-1}, t), \; f^{\psi_A}(\mathbf{X}_{k-1}, \mathbf{A}_{k-1}, t)$;        $\triangleright$ First drift evaluation
5:     **if** $k > 1$ **then**
6:         Compute $V_{\mathbf{X},k}, V_{\mathbf{A},k}$ using $(\boldsymbol{f}_{\mathbf{X},k}^{(1)}, \boldsymbol{f}_{\mathbf{X},k-1}^{(1)})$ and $(\boldsymbol{f}_{\mathbf{A},k}^{(1)}, \boldsymbol{f}_{\mathbf{A},k-1}^{(1)})$;        $\triangleright$ Eq. (13)
7:         Update $\overline{V}_{\mathbf{X}}, \overline{V}_{\mathbf{A}}$ by $V_{\mathbf{X},k}, V_{\mathbf{A},k}$;                         $\triangleright$ Eq. (14)
8:         Compute $\Delta t_{\mathbf{X},k}, \Delta t_{\mathbf{A},k}$ and set $\Delta t_k \leftarrow \min(\Delta t_{\mathbf{X},k}, \Delta t_{\mathbf{A},k})$;          $\triangleright$ Eq. (15)
9:         $\overline{V}_{\mathbf{X}}, \overline{V}_{\mathbf{A}} \leftarrow \gamma \cdot (\overline{V}_{\mathbf{X}} + \overline{V}_{\mathbf{A}})$;                       $\triangleright$ Update variation state
10:     **else**
11:         $\Delta t_k \leftarrow \Delta t_{\text{base}}$;
12:     **end if**
13:     $\Delta t_k \leftarrow \min(\Delta t_k, T - t)$;
14:     $\hat{\mathbf{X}}_k \leftarrow \mathbf{X}_{k-1} + \boldsymbol{f}_{\mathbf{X},k}^{(1)}\Delta t_k + g_k\sqrt{\Delta t_k}\mathbf{Z}_{\mathbf{X}}$;                  $\triangleright$ Predict intermediate state
15:     $\hat{\mathbf{A}}_k \leftarrow \mathbf{A}_{k-1} + \boldsymbol{f}_{\mathbf{A},k}^{(1)}\Delta t_k + g_k\sqrt{\Delta t_k}\mathbf{Z}_{\mathbf{A}}$;
16:     $\boldsymbol{f}_{\mathbf{X},k}^{(2)}, \boldsymbol{f}_{\mathbf{A},k}^{(2)} \leftarrow f^{\psi_X}(\hat{\mathbf{X}}_k, \hat{\mathbf{A}}_k, t + \Delta t_k), \; f^{\psi_A}(\hat{\mathbf{X}}_k, \hat{\mathbf{A}}_k, t + \Delta t_k)$;    $\triangleright$ Second drift evaluation
17:     $\bar{\boldsymbol{f}}_{\mathbf{X},k} \leftarrow \frac{1}{2}(\boldsymbol{f}_{\mathbf{X},k}^{(1)} + \boldsymbol{f}_{\mathbf{X},k}^{(2)}), \quad \bar{\boldsymbol{f}}_{\mathbf{A},k} \leftarrow \frac{1}{2}(\boldsymbol{f}_{\mathbf{A},k}^{(1)} + \boldsymbol{f}_{\mathbf{A},k}^{(2)})$;
18:     $\mathbf{Z}_{\mathbf{X}}, \mathbf{Z}_{\mathbf{A}} \sim \mathcal{N}(\mathbf{0}, \mathbf{I})$;
19:     $\mathbf{X}_k \leftarrow \mathbf{X}_{k-1} + \bar{\boldsymbol{f}}_{\mathbf{X},k}\Delta t_k + g_k\sqrt{\Delta t_k}\mathbf{Z}_{\mathbf{X}}$;                   $\triangleright$ Second-order update
20:     $\mathbf{A}_k \leftarrow \mathbf{A}_{k-1} + \bar{\boldsymbol{f}}_{\mathbf{A},k}\Delta t_k + g_k\sqrt{\Delta t_k}\mathbf{Z}_{\mathbf{A}}$;
21:     $t \leftarrow t + \Delta t_k, \quad k \leftarrow k + 1$;
22: **end while**
23: **Output:** A graph $(\mathbf{X}_T, \mathbf{A}_T)$.

---

*Table 9.* **Statistical significance analysis on QM9 using the GDSS model.** The "Diff." column denotes the performance difference between DVS and the fixed-step Euler-Maruyama sampler.

| Metric | DVS (Ours) | Fixed-Step | Diff. | 95% CI of Diff. | $p$-value | Significance |
|---|---|---|---|---|---|---|
| **Valid (%)** $\uparrow$ | $95.40 \pm 0.25$ | $95.18 \pm 0.19$ | $+0.21 \pm 0.28$ | $[0.01, 0.41]$ | $0.040$ | $p < 0.05$ |
| **FCD** $\downarrow$ | $2.472 \pm 0.040$ | $2.576 \pm 0.044$ | $-0.104 \pm 0.017$ | $[-0.12, -0.09]$ | $< 0.001$ | $p < 0.001$ |
| **NSPDK** $\downarrow$ | $0.0033 \pm 0.0001$ | $0.0034 \pm 0.0001$ | $-0.0001 \pm 0.0001$ | $[-0.0001, -0.0000]$ | $0.011$ | $p < 0.05$ |

*Table 10.* Runtime comparison between fixed-step Euler-Maruyama and DVS-driven sampling.

| Dataset | Method | Time per Step (s) | NFE ($K$) | Total Time (s) |
|---|---|---|---|---|
| QM9 | Euler-Maruyama (Fixed-Step) | 0.843 | 1000 | 843 |
| | **DVS-Euler-Maruyama (Ours)** | **0.898 (+6.53%)** | **755** | **678 (-19.57%)** |
| ZINC250k | Euler-Maruyama (Fixed-Step) | 5.695 | 1000 | 5695 |
| | **DVS-Euler-Maruyama (Ours)** | **5.700 (+0.09%)** | **1018** | **5803 (+1.90%)** |

operations with $\mathcal{O}(N^2)$ scaling. For example, in Graph Transformer backbones, a single forward pass has complexity $\mathcal{O}(L \cdot N^2 \cdot D_h)$, where $L$ is the number of layers, $N$ is the number of nodes, and $D_h$ is the hidden dimension.

In contrast, DVS only computes the difference between consecutive drift evaluations, followed by an $\ell_2$ norm and simple scalar operations, incurring an additional $\mathcal{O}(D)$ cost per step, where $D$ is the dimensionality of the state. Since this overhead is negligible compared to $C_f$ in practical settings, the overall asymptotic complexity remains unchanged.

To validate the analysis, we measure the wall-clock sampling time with and without DVS on the GruM model for QM9 and ZINC250k (see *Table 10*). The results show that the per-step overhead introduced by DVS is minimal (6.53% on QM9 and 0.09% on ZINC250k). Despite this small overhead, DVS reduces the total runtime by 19.57% on QM9 due to fewer function evaluations, while maintaining comparable efficiency on ZINC250k (+1.90%). These results confirm that DVS introduces negligible computational overhead and achieves improved or similar overall efficiency depending on the task.

# D. Visualizations of Generated Samples

In this section, we provide qualitative visualizations to demonstrate the effectiveness of the DVS-driven adaptive sampler. We analyze both the final generation quality for molecular datasets and the intermediate structural evolution for synthetic graph datasets.

## D.1. Molecular Graph Generation

*Figure 5* displays the 3D ball-and-stick models of molecules generated by the GruM model equipped with our DVS-Euler-Maruyama sampler.

The first row of *Figure 5* shows samples from the QM9 dataset. Despite the small size of these molecules, the DVS sampler ensures precise bond lengths and angles, resulting in chemically plausible organic structures. The second row showcases ZINC250k samples, which are significantly more complex. Our sampler successfully resolves intricate drug-like motifs, including fused rings, long-chain substituents, and various heteroatoms (represented by distinct colors). The high visual quality and structural diversity of these samples indicate that achieving an arc-length parametrization allows the model to better navigate the complex chemical manifold.

## D.2. Evolution of Graph Structures

To compare the sampling dynamics, we visualize the snapshots of the reverse-time generation process at intervals $t \in \{0, 0.2, 0.4, 0.6, 0.8, 1.0\}$. *Figures 6* and *7* illustrate the evolution for Planar and SBM datasets, respectively.

In *Figure 6* (Planar), both samplers begin with a dense, noisy edge distribution ($t = 0$). In the fixed-step trajectory (top row), the transition from noise to a sparse skeleton is relatively abrupt. In contrast, the DVS trajectory (bottom row) demonstrates a more meticulous refinement. By $t = 0.6$ and $0.8$, the DVS sampler has already significantly reduced "noisy" overlapping

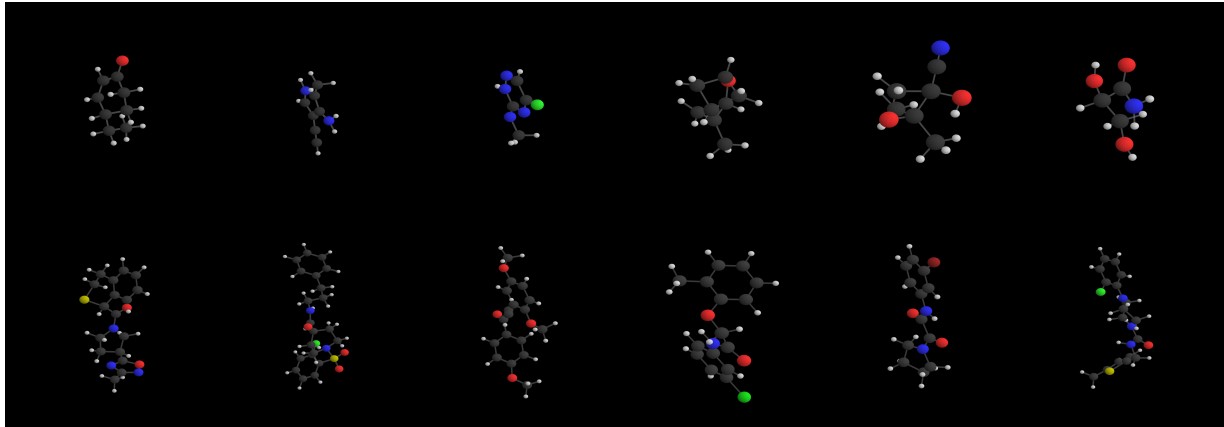

*Figure 5.* **Visualizations of generated molecules on QM9 (top row) and ZINC250k (bottom row).** The samples are generated using the GruM model with the DVS-Euler-Maruyama sampler. The top row shows organic molecules from QM9 with realistic geometric configurations. The bottom row displays complex drug-like molecules from ZINC250k, successfully capturing multi-ring systems and diverse heteroatoms (e.g., O, N, S, Cl).

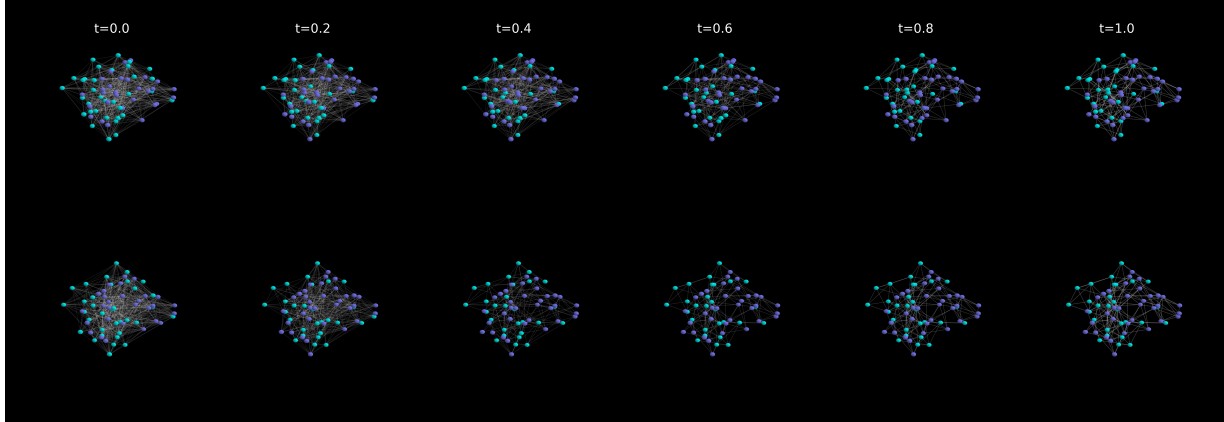

*Figure 6.* **Snapshots of structural evolution on the Planar dataset.** The **top row** represents the standard fixed-step Euler sampler, while the **bottom row** represents our DVS-Euler-Maruyama sampler. As $t$ increases, the DVS-driven sampler resolves the sparse planar skeleton more cleanly, especially in the late stages ($t \geq 0.6$).

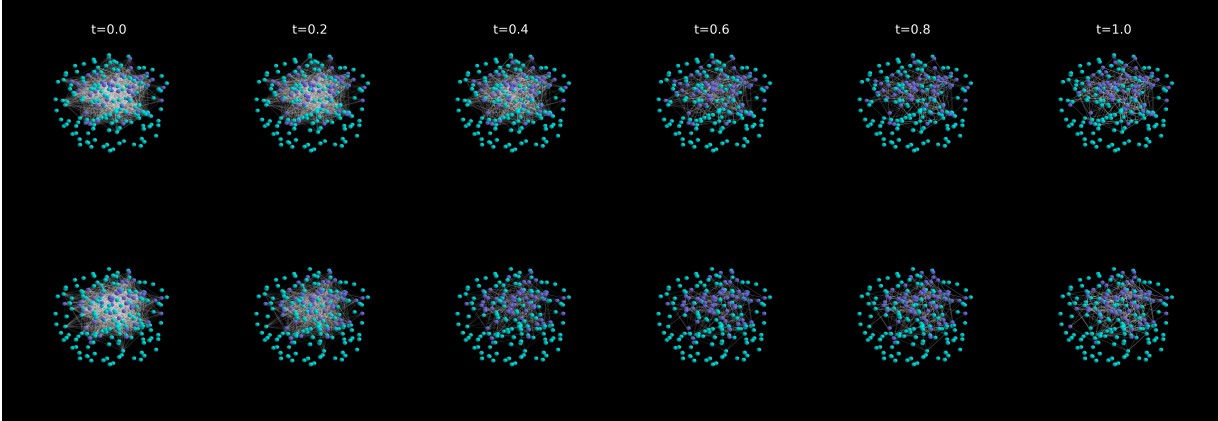

*Figure 7.* **Snapshots of structural evolution on the SBM dataset.** The **top row** (fixed-step) and **bottom row** (DVS-Euler-Maruyama) both start from a dense noise state. However, the DVS-driven sampler achieves clearer community separation and fewer spurious inter-cluster edges by the final step ($t = 1.0$).

edges, leading to a much cleaner and more accurate planar grid at $t = 1.0$.

Similarly, in *Figure* 7 (SBM), the DVS-driven sampler (bottom row) shows a more controlled community formation process. While the fixed-step baseline (top row) retains a noticeable amount of unnecessary edges around the clusters even at $t = 0.8$, the DVS effectively suppresses these spurious connections during the stiff crystallization phase. This results in a final state ($t = 1.0$) where the three community clusters are more distinct and better organized. These results confirm that DVS's ability to adapt step sizes based on local curvature is crucial for capturing both global skeletons and local community structures.

