# OpenReview forum: "Information-Geometric Adaptive Sampling for Graph Diffusion"
_ICML.cc/2026/Conference — ICML 2026 regular_

### Official Review · Reviewer_iJak · 2026-02-23

**Soundness:** 3
**Presentation:** 2
**Significance:** 3
**Originality:** 3
**Overall Recommendation:** 5
**Confidence:** 4

**Summary:**

This paper investigates the adaptive adjustment of time steps in diffusion models for graph generation. The authors introduce the concept of DVS based on information geometry (statistical manifold) and subsequently propose an information-geometric adaptive sampling method for graph diffusion models. Compared to existing fixed-step or heuristic sampling approaches, DVS sampling provides a geometrically grounded interpretation of the diffusion trajectory. Correspondingly, the authors conduct experiments on multiple datasets, including QM9, ZINC250k, Planar, SBM and Ego-small, and experimental results demonstrate that DVS sampling serves as an effective approach for improving both generation quality and efficiency.

**Compliance With Llm Reviewing Policy:**

Affirmed.

**Final Justification:**

The authors have addressed my main concerns. Accordingly, I maintain my overall recommendation.

**Key Questions For Authors:**

**Q1.** In **Sec. 3.2**, the authors note that, based on experimental observations, the variation in noise scale along the sampling trajectory is several orders of magnitude smaller than that of the drift field, and therefore treat it as approximately constant. However, this difference in magnitude may be dataset-dependent. If the authors cannot provide a clear theoretical justification, this dependency should be explicitly stated in the main text.

**Q2.** Throughout the paper, the authors emphasize that their adaptive strategy allows the DVS sampler to maintain a constant informational distance per step (e.g., in the list of contributions). However, keeping this distance constant is not a necessary choice. Given this geometric insight, one might consider adjusting the informational distance at each step based on the underlying geometry, rather than fixing it, as a potential approach to further enhance both efficiency and generation quality. Is the authors' current design motivated by a desire to preserve model simplicity, or are there other underlying reasons?

**Q3.** In the list of contributions, the authors state that incorporating DVS sampling introduces only negligible computational overhead. However, no specific evidence or elaboration supporting this claim is provided (including in **Appendix**). It would be helpful if the authors could include a comparative analysis of computational complexity and report the empirical computation time with and without DVS sampling in their experiments.

**Q4.** In the ablation study presented in **Sec. 4.4**, the authors note that a larger $\tau$ results in a more conservative sampling strategy, a finding that is clearly supported by the QM9 experiments. However, it appears that this relationship may not be universally consistent. If the DVS (or curvature) along the trajectory is monotonically increasing or decreasing within a certain segment, a larger $\tau$ could respectively lead to a more conservative or more aggressive strategy, rather than uniformly the former. This suggests a potential dependence on the dataset. Could the authors provide theoretical justification for their original claim?

**Q5.** Regarding the references, I would like to point out the following issues. The authors should carefully check all references based on these problems and make corrections.

The citations "Bortoli et al., 2021" and "De Bortoli et al., 2021" refer to the same paper but are listed as two separate entries and cited together. This is a fundamental error that should be corrected.

The closely related model GruM is inconsistently referred to as both "GRUM" and "GruM" throughout the main text and **Appendix**. A unified notation (preferably "GruM") should be adopted.

The reference formatting is inconsistent and should be carefully revised.

Many entries fail to capitalize the first letters of words where required (e.g., schr\"{o} dinger,  langevin and hamiltonian monte carlo methods, Fisher-rao, ornstein-uhlenbeck), which needs to be addressed systematically.

**Limitations:**

Yes

**Strengths And Weaknesses:**

**Strengths:**
1. This paper appropriately leverages the geometric structure of statistical manifolds to elucidate the trajectory sampling mechanism in graph generation diffusion models, and the theoretical exposition of the core idea is clear.
2. Regarding the proposed DVS sampling method, the authors have conducted extensive experiments. The empirical results substantiate the theoretical concept, and the performance of the method is satisfactory.

**Weaknesses:**
1. Leveraging geometric structures to guide sampling is a natural idea, and the theoretical portion of the paper is generally clear. However, several points remain unclear (see **Q1-2**).
2. Regarding the experiments, a few specific details require clarification from the authors (see **Q3-4**).
3. The main text of the paper is well-written; however, the references contain both citation errors and numerous formatting issues. The authors must carefully revise these (I have provided examples in **Q5**).

---

> ### Author Rebuttal · Authors · 2026-03-31
>
> We sincerely appreciate your constructive feedback and insightful comments. Our point-by-point responses are provided below.
>
> ### **Response to Q1: Drift vs. Noise Scale Variation**
> While the statement in $\S3.2$ is consistent with our empirical observations, it follows from a general scaling property of the underlying SDE rather than being dataset-dependent. The noise schedule $\sigma(t)$ is a deterministic and smooth function of time, whose infinitesimal variation satisfies $|d\sigma| = |\dot{\sigma}(t)| dt$ with finite $\dot{\sigma}(t)$, implying $|d\sigma| = O(dt)$. In contrast, the drift field $\mathbf{f}(\mathbf{x}, t)$ evolves along a stochastic trajectory, and by Itō’s lemma its leading-order variation is dominated by the stochastic term $d\mathbf{f} \approx \sigma \mathbf{J} _ {\mathbf{f}} d\mathbf{W} + O(dt)$, where $\mathbf{J} _ {\mathbf{f}} = \nabla_{\mathbf{x}} \mathbf{f}$ and $W(t)$ is a standard Wiener process. Since $d\mathbf{W} \sim O(\sqrt{dt})$, this yields $\Vert d\mathbf{f} \Vert_2 = O(\sqrt{dt})$. Therefore, the ratio $\Vert d\mathbf{f} \Vert_2 / |d\sigma| \approx (\sigma \sqrt{\mathrm{Tr}(\mathbf{J} _ {\mathbf{f}}^\top \mathbf{J} _ {\mathbf{f}})} / |\dot{\sigma}|) \cdot (1/\sqrt{dt}) \propto 1/\sqrt{dt} \to \infty$ as $dt \to 0$, showing that the dominance of drift variation over noise-scale variation is a structural consequence of the SDE. Consequently, treating $\sigma(t)$ as approximately constant within each step is theoretically justified.
>
> ### **Response to Q2: Constant Information Increments**
> The core insight of DVS is to enforce a uniform rate of change in the probability distribution along the sampling trajectory. In other words, it moves at a uniform informational speed along the manifold. This design ensures adaptive fidelity by preventing the solver from skipping over critical structural details in high-variation regions, while simultaneously accelerating through smoother regions to eliminate redundant steps. Unlike alternative schemes that rely on expensive higher-order geometric quantities such as curvature or Hessians, DVS achieves geometry-aware sampling using only drift variations, resulting in a simple and efficient implementation.
>
> ### **Response to Q3: Complexity**
> DVS is a lightweight mechanism that introduces no additional neural network evaluations. The dominant cost of diffusion sampling lies in evaluating the drift network $f_t(x)$, resulting in a total complexity of $\mathcal{O}(K \cdot C_f)$, where $K$ is the number of steps and $C_f$ is the cost of one forward pass. For graph diffusion models, this cost is typically dominated by dense operations with $\mathcal{O}(N^2)$ scaling. For instance, in Graph Transformer backbones, a single forward pass has complexity $\mathcal{O}(L \cdot N^2 \cdot d_h)$, where $L$ is the number of layers, $N$ is the number of nodes, and $d_h$ is the hidden dimension. In contrast, DVS only computes the difference between consecutive drift evaluations followed by an $\ell_2$ norm and simple scalar operations, incurring an additional $\mathcal{O}(D)$ cost per step, where $D$ is the dimensionality of the state. Since this cost is negligible compared to $C_f$ in practical settings, the overall complexity remains unchanged.
>
> | Dataset | Method | Time per Step (s) | Total NFE ($K$) | Total Runtime (s) |
> | :--- | :--- | :---: | :---: | :---: |
> | **QM9** | Euler-Maruyama (Fixed-Step) | 0.843 | 1000 | 843 |
> | | **DVS-Euler (Ours)** | **0.898 (+6.53%)** | **755** | **678 (-19.57%)** |
> | **ZINC250k** | Euler-Maruyama (Fixed-Step) | 5.695 | 1000 | 5695 |
> | | **DVS-Euler (Ours)** | **5.700 (+0.09%)** | **1018** | **5803 (+1.90%)** |
>
> To support the theoretical analysis, we report the average wall-clock sampling time with and without DVS on the GruM model for QM9 and ZINC250k. The results show that the per-step overhead of DVS is minimal (6.53% on QM9 and 0.09% on ZINC250k). Despite this, DVS reduces total runtime by 19.57% on QM9 due to fewer function evaluations, while maintaining comparable efficiency on ZINC250k. These findings confirm that DVS introduces negligible overhead and can achieve improved or similar overall efficiency.
>
> ### **Response to Q4: Effect of $\tau$**
> The observation that a larger $\tau$ leads to more conservative sampling follows from the design of our update rule. Specifically, $\tau$ is an aggregation factor for the node and edge variation signals, and a larger $\tau$ leads to a higher overall level of the EMA-smoothed DVS. Since the step size follows an inverse power-law relation $\Delta t _ k \propto (1/\text{DVS} _ k)^{\beta}$, this increase in DVS directly leads to smaller step sizes, effectively shrinking them across the trajectory. This explains why the conservative effect of a larger $\tau$ is consistently observed across different datasets.
>
> ### **Response to Q5: Reference and Formatting**
> All reference inconsistencies, duplicate entries, naming issues, and formatting errors have been carefully corrected.

---

> > ### Author Rebuttal · Reviewer_iJak · 2026-04-01
> >
> > Except for **Q4**, the authors’ response is satisfactory. Regarding **Q4**, I would like to clarify my concern in a more specific way. For example, along the trajectory, DVS as a continuous function of time has a derivative with respect to $t$ that changes from being extremely close to $0$ to becoming very large. According to **Eq. (14)** and **Algorithm 1**, with $\tau_1=0.1$ and $\tau_2=10^3$, I still believe that in this case $\tau_2$ may lead to more aggressive sampling (compared to $\tau_1$).
> >
> > Regarding **Q2**, after the main idea is established, it is still necessary to clarify every specific detail in the framework before experiments and downstream applications can proceed. At this stage, multiple reasonable design choices exist. For instance, the authors opt to maintain a constant informational distance per step. My original question was whether there might be any deeper rationale behind this choice that I had overlooked. The authors' response that "resulting in a simple and efficient implementation", sufficiently clarifies the matter. That said, if one were to design the architecture from a purely geometric standpoint, a varying informational distance per step would naturally extract more benefit from the underlying geometry, though potentially at the expense of computational efficiency or architectural simplicity. In any case, this remains an interesting direction to explore in future work.

---

> > > ### Author Response · Authors · 2026-04-03
> > >
> > > **Response to Q4:**
> > >
> > > > Let $\overline{V}$ denote the EMA‑smoothed DVS and $V$ denote the the instantaneous DVS. Our algorithm’s core logic can be expressed in three lines:
> > > >
> > > > 1. **Smoothing (Eq. 14 / Algorithm 1 line 5):** $\overline{V} \leftarrow (1-\alpha)\overline{V} + \alpha V$  (only suppresses noise in $V$)
> > > > 2. **Step size (Eq. 15 / Algorithm 1 line 6):** $\Delta t \propto 1/\overline{V}$  (from $\Delta s^2 = \overline{V}\Delta t \approx \text{const}$)
> > > > 3. **Memory scaling (Algorithm  1 line 8):** $\overline{V} \leftarrow \tau \cdot \overline{V}$  (after computing DVS, the memory is scaled by $\tau$)
> > > >
> > > > **Key observations:**
> > > > - We only use **the smoothed** $\overline{V}$ to tune the step size $\Delta t$, never use the instantaneous $V$.
> > > > - Because $\tau>0$ multiplies $\overline{V}$ directly, a **larger $\tau$ makes $\overline{V}$ larger** at every step.
> > > > - From $\Delta t \propto 1/\overline{V}$, larger $\overline{V}$ forces **smaller $\Delta t$** → more conservative sampling.
> > >
> > > The EMA formula serves a singular purpose: to act as a temporal low-pass filter that smooths out stochastic noise or anomalous outliers in the instantaneous $V$ as it evolves overtime. Our step-size controller relies entirely on the smoothed state $\overline{V}$. The instantaneous $V$ is never used directly to dictate $\Delta t$. Hence a larger $\tau$ always yields smaller steps, as confirmed in Table 5.

---

### Official Review · Reviewer_JLwk · 2026-02-28

**Soundness:** 3
**Presentation:** 3
**Significance:** 2
**Originality:** 2
**Overall Recommendation:** 5
**Confidence:** 2

**Summary:**

The paper “ Information-Geometric Adaptive Sampling for Graph Diffusion“ presents an adaptive sampling strategy for diffusion models based on notions from information geometry. More specifically, the integration time step for the diffusive backward SDE is estimated directly from the Fisher-Rao arc length of the model. This approach offers an improvement with respect to Fixed-Step (Jo et al., 2024) sampling method because: on one hand, a fixed time step might imply overshooting at small diffusion times, since the model is more susceptible to perturbations of the score; on the other hand, the same fixed step might slow down sampling at large times, where distributions evolve slowly.

**Compliance With Llm Reviewing Policy:**

Affirmed.

**Key Questions For Authors:**

I will now report questions and comments that I am asking the Authors to address, in order to improve their manuscript and drive towards my evaluation of the manuscript:

1. Graph Diffusion Models are indeed an interesting problem and the paper presents a useful tool for sampling graphs. Since the method looks very general, I ask the authors to comment on whether and how the same informed sampling method could be employed on more standard data-sets learned by diffusion models, say image data-sets, to speed up sampling. Moreover, the paper “Fisher Information Improved Training-Free Conditional Diffusion Model” by Song & Lai (2024) proposes a similar type of reasoning for estimating guided scores in diffusion models. I ask the authors to discuss similarities and differences, and eventually suggest to cite this paper in the Related Work, given the relevance.
2. It can be inferred from the paper that the models employed in experiments must be trained to learn a score function defined on continuous times, instead of DDPMs. Nevertheless this aspect is poorly addressed both in the manuscript that, reasonably, spends most of the space to discuss generation, and in the appendices. I would ask the Authors to at least mention the way the score f_t is learned by the models that can make use of DVS sampling method.
3. Equation 15 is not fully clear to me. How is the clip() function defined? Could the authors comment a bit more extensively on the choice of such function for determining the adapted time step?
4. Figure 3 represents a comparison between the Euler-Maruyama integrated SDE with DVS (red) and the Euler method using a fixed-step size (blue). I was wondering why Euler was employed, to show the fixed step size results, instead of the same Euler-Maruyama integration algorithm.
5. Still around paragraph “Arc-Length Analysis” : the information increment across the two methods in Figure 3 still follow similar trends, even though the predicted effect is clearly visible from the absence of the final explosion in the red line. I was wondering whether ds^2 could really be constant along time if one adapted, or even learned, the hyper-parameters k_ref and \beta in Eq. (15).

**Limitations:**

Yes.

**Strengths And Weaknesses:**

**Strengths**:

The generation method proposed by the paper seems novel to me because it relies on an estimate of the "real" Fisher information of a diffusion model. In fact, most of the papers in the literature address to this quantity as the Jacobian of the (Stein) score function, i.e. the second derivative of the log-likelihood with respect to the space coordinates x. This quantity gives useful geometric insights about the score function vector field, but it is not related to the actual Fisher information metrics, which instead represents the susceptibility of the model under perturbations of the parameters.
Furthermore, I retain the presentation of the manuscript clear, even for non experts in information geometry.

**Weaknesses**:

Despite the general derivation of the Fisher-Rao arc length, the paper does not motivate in depth the application to graph generation, and how this method should suite this implementation more than others.

---

> ### Author Rebuttal · Authors · 2026-03-31
>
> We sincerely appreciate your constructive feedback and insightful comments. Our point-by-point responses are provided below.
>
> ### **Response to Q1: Generality and Comparison with Prior Work**
>
> Our method operates at the level of the reverse SDE solver and is independent of the data modality. It estimates local variation intensity through drift changes combined with the noise scale, both of which are available in standard continuous-time diffusion models, and can therefore be directly applied to image diffusion models without modification. Notably, it is particularly effective for graph generation, where the data is discrete and heterogeneous, and the dynamics involve abrupt structural changes such as connectivity or node/edge variations. In such settings, DVS naturally allocates finer steps to these high-variation regions, enabling more accurate modeling.
>
> Regarding Song & Lai (2024), both approaches leverage Fisher information to improve diffusion sampling, and we will include it in the related work. However, they differ in their level of intervention. Their method operates on the score function, reweighting it via the Cramér–Rao bound to modify the sampling direction, whereas our DVS framework adaptively adjusts the timestep $\Delta t$ without altering the underlying dynamics. As a result, the two methods are complementary: theirs improves the dynamics, while ours improves the discretization, and they can be naturally combined.
>
> ### **Response to Q2: Reverse Drift Construction**
>
> Our DVS sampler requires access to the continuous-time reverse drift $f_t(x)$ along the sampling trajectory. In score-based SDE models such as VP-SDE and VE-SDE, $f_t$ is constructed from a time-dependent score network $s_\theta(x,t)$ trained via denoising score matching. For example, in VP-SDE, the reverse drift and diffusion take the form $f_t(x) = -\tfrac{1}{2}\beta_t x - \beta_t s_\theta(x,t)$ and $\sigma_t = \sqrt{\beta_t}$; in VE-SDE, $f_t(x) = -\sigma_t^2 s_\theta(x,t)$ with diffusion coefficient $\sigma_t$. Beyond score-based models, frameworks such as GruM also define a continuous-time drift, where $f_t(G_t) = \alpha \sigma_t^2 G_t + \frac{\sigma_t^2}{v_t}\left(\frac{s_\theta(G_t,t)}{u_t} - G_t\right)$ and the diffusion term is $\sigma_t dW_t$, with $s_\theta$ predicting the target graph structure and the drift driving the trajectory toward it. Despite these different parameterizations, all these models provide a well-defined $f_t$ along the trajectory, which DVS uses to estimate local variation and adapt step sizes without modifying the training procedure.
>
> ### **Response to Q3: Step-Size Scaling and Clipping**
>
> In our framework, the adaptive step size is computed as $\Delta t_k^{\text{raw}} = \Delta t_{\text{base}} \left( \frac{\kappa_{\mathrm{ref}}}{\text{DVS}_k} \right)^\beta$. This power-law form reflects an inverse dependence on drift variation: regions with larger variation lead to smaller steps to preserve sampling fidelity, while smoother regions allow larger steps to improve efficiency. The exponent $\beta$ modulates the sensitivity of this adjustment.
>
> The clipping operator $\text{clip}(x,a,b)=\min(\max(x,a),b)$ restricts a value to the interval $[a,b]$. In our method, due to local variability in $\text{DVS} _ k$, the computed step size may occasionally become too large or too small, so we apply $\Delta t _ k = \text{clip}(\Delta t _ k^{\text{raw}}, \Delta t_{\min}, \Delta t_{\max})$ to enforce reasonable bounds and ensure stable integration. This clipping only acts as a safeguard in extreme cases.
>
> ### **Response to Q4: Euler–Maruyama Clarification**
>
> We used the term “Euler” in Fig. 3 as a shorthand for Euler–Maruyama, which may cause confusion since Euler and Euler–Maruyama correspond to ODEs and SDEs respectively. In our setting, the reverse process is modeled as an SDE, and the baseline in Fig. 3 indeed uses Euler–Maruyama discretization with a fixed step size. We will revise the figure caption to explicitly state “Euler–Maruyama (fixed step)” to remove this ambiguity.
>
> ### **Response to Q5: Informational Increments and Parameter Adaptation**
>
> Regarding Figure 3, DVS fundamentally reconfigures the allocation of informational progress across the entire trajectory rather than just mitigating the final explosion. While the baseline takes unnecessarily small steps in early stages, leading to negligible informational progress, the DVS sampler actively increases the step size in these low-variation regions to make more effective use of the sampling budget.
>
> In principle, further tuning or learning $\kappa_{\mathrm{ref}}$ and $\beta$ could better enforce uniform arc-length increments, consistent with the theoretical relation $\Delta t _ k \propto 1 / \text{DVS} _ k$. Empirically, we observe that larger graph datasets tend to require larger $\kappa_{\mathrm{ref}}$ due to increased structural complexity, and we consider systematic parameter learning an important direction for future work.

---

> > ### Author Rebuttal · Reviewer_JLwk · 2026-04-02
> >
> > I thank the Authors of the paper for addressing all my questions in detail.
> >
> > I am happy with the response and I will keep my current score.

---

### Official Review · Reviewer_a7Np · 2026-03-13

**Soundness:** 2
**Presentation:** 3
**Significance:** 3
**Originality:** 2
**Overall Recommendation:** 4
**Confidence:** 2

**Summary:**

This paper addresses the inefficiency of uniform time-stepping in graph diffusion model sampling. The authors propose treating the sequence of transition kernels as a curve on a Riemannian statistical manifold equipped with the Fisher-Rao metric, deriving a Drift Variation Score that measures normalized drift change rate, and using it to adaptively control step sizes so that each step covers approximately equal information-geometric arc length. On the theoretical side, the paper shows that the Fisher information matrix for the Gaussian transition kernel is proportional to $\frac{dt}{\sigma_t^2}I$, which reduces the arc-length element to a simple ratio of squared drift variation over noise variance, and argues empirically that the noise-schedule contribution is negligible. Experiments on molecular and general graph benchmarks using GRUM and GDSS models show consistent improvements in FCD and MMD metrics over fixed-step and quadratic schedules, while using fewer sampling steps.

**Compliance With Llm Reviewing Policy:**

Affirmed.

**Final Justification:**

Since the authors have addressed my concerns, I am positive about this paper.

**Key Questions For Authors:**

All reported improvements lack error bars. Given that graph generation metrics such as FCD are known to exhibit non-trivial variance across random seeds, could the authors provide results over multiple independent runs (e.g., 3-5 seeds) with standard deviations?

**Limitations:**

yes

**Strengths And Weaknesses:**

Strength:

- Combining the Fisher-Rao metric with diffusion sampling to derive a training-free, plug-and-play step-size controller for graph generation is a novel and conceptually clean idea.

- The paper has a clear narrative arc from motivation to method to experiments, and Figure 3's arc-length analysis is a particularly effective visualization.

- The consistent finding that DVS-Euler rivals or surpasses Fixed-Step Heun across multiple benchmarks suggests that step allocation strategy may matter more than integrator order for graph-structured data, which is a practically useful insight for the community.

Weakness:

- the manually specified per-dataset active ranges in Table 7 contradict the claim that the method automatically senses local geometry without prior knowledge.

---

> ### Author Rebuttal · Authors · 2026-03-31
>
> We sincerely appreciate your constructive feedback and insightful comments. Our point-by-point responses are provided below.
>
> ### **Response to Concern on Error Bars**
>
> To evaluate the impact of randomness, we conduct experiments comparing the Fixed-Step Euler–Maruyama sampler with the DVS-driven Euler–Maruyama sampler on the QM9 dataset with GDSS, as well as on QM9 and ZINC250k datasets with GruM. For each setting, we ran 5 independent trials using the same set of random seeds for both samplers, while keeping all configurations consistent with those described in the paper. We report the mean ± standard deviation for all metrics, and perform paired t-tests to assess statistical significance. The results are summarized in the table below.
>
> | Metric | Method | GDSS (QM9) | GruM (QM9) | GruM (ZINC250k) |
> | :--- | :--- | :---: | :---: | :---: |
> | **Valid (%) ↑** | Fixed-Step | $95.12 \pm 0.17$ | $99.44 \pm 0.07$ | $98.37 \pm 0.13$ |
> | | **DVS** | $\mathbf{95.41 \pm 0.31}$ | $\mathbf{99.49 \pm 0.04}$ | $\mathbf{98.47 \pm 0.08}$ |
> | **FCD ↓** | Fixed-Step | $2.554 \pm 0.028$ | $0.1205 \pm 0.0084$ | $2.226 \pm 0.034$ |
> | | **DVS** | $\mathbf{2.442 \pm 0.024}$ | $\mathbf{0.1095 \pm 0.0094}$ | $\mathbf{2.144 \pm 0.032}$ |
> | **NSPDK ↓** | Fixed-Step | $\mathbf{0.0034 \pm 0.0001}$ | $0.0003 \pm 0.0001$ | $0.0017 \pm 0.0001$ |
> | | **DVS** | $\mathbf{0.0034 \pm 0.0001}$ | $\mathbf{0.0002 \pm 0.0001}$ | $\mathbf{0.0016 \pm 0.0001}$ |
> | **$p$-value** | (FCD) | **< 0.001** | **< 0.001** | **< 0.001** |
>
>
> The results show that DVS consistently improves FCD across all datasets and model backbones, with all improvements being highly statistically significant (p < 0.001), demonstrating that the gains are robust despite the known variance of FCD. Meanwhile, validity remains consistently high, and NSPDK shows slight improvements across all settings, indicating that DVS not only preserves structural properties but also provides additional gains. Overall, these results demonstrate that DVS reliably enhances distributional quality while simultaneously improving or maintaining other metrics, yielding a consistent advantage over the Fixed-Step Euler–Maruyama baseline.
>
>
> ### **Response to Concern on Active Ranges**
>
> The active range can be viewed as an optional design choice rather than an essential component of DVS. It serves as a tunable hyperparameter that governs the trade-off between computational efficiency and performance, and it can be tuned on a validation set. Importantly, introducing such a range does not alter the underlying geometry-driven mechanism of DVS. The method remains fundamentally general and prior-free, with full-trajectory adaptation serving as the default formulation. The results presented in the table below provide empirical evidence corroborating this claim, showing that DVS can be directly applied over the entire sampling trajectory without manual restriction. On the Planar and SBM datasets using the GruM model, full-trajectory DVS consistently outperforms the fixed-step Euler–Maruyama baseline across all metrics, demonstrating its inherent stability and effectiveness even in the absence of specific range selection.
>
> | Dataset | Method | Deg. $\downarrow$ | Clus. $\downarrow$ | Orbit $\downarrow$ | Spec. $\downarrow$ |
> | :--- | :--- | :---: | :---: | :---: | :---: |
> | **Planar** | Fixed-Step | 0.0002 | 0.0300 | 0.0013 | 0.0060 |
> | | Full DVS  | 0.0001 | 0.0294 | 0.0013 | 0.0058 |
> | | Active-Range DVS | 0.0001 | 0.0283 | 0.0010 | 0.0052 |
> | **SBM** | Fixed-Step | 0.0006 | 0.0498 | 0.0455 | 0.0051 |
> | | Full DVS | 0.0002 | 0.0481 | 0.0386 | 0.0049 |
> | | Active-Range DVS | 0.0007 | 0.0477 | 0.0382 | 0.0030 |

---

> > ### Author Rebuttal · Reviewer_a7Np · 2026-04-04
> >
> > Since the authors have addressed my concerns, I am positive about this paper.

---

### Official Review · Reviewer_VfSh · 2026-03-15

**Soundness:** 2
**Presentation:** 2
**Significance:** 3
**Originality:** 3
**Overall Recommendation:** 4
**Confidence:** 4

**Summary:**

This paper proposes an information-geometric framework for adaptive time-stepping in graph diffusion models. The authors reinterpret the reverse-time SDE sampling trajectory as a parametric curve on a Riemannian statistical manifold equipped with the Fisher-Rao metric. From this perspective, they derive the Drift Variation Score (DVS), a training-free metric that quantifies the instantaneous rate of informational change. The DVS-driven adaptive sampler dynamically adjusts step sizes to maintain approximately constant informational progress per step -- shrinking steps in high-curvature regions and expanding them in stable regimes. Experiments on molecular and social network graph generation benchmarks show modest improvements over fixed-step and quadratic schedules.

**Compliance With Llm Reviewing Policy:**

Affirmed.

**Final Justification:**

The detailed responses given in the rebuttal should be incorporated into the final version of the paper.

**Key Questions For Authors:**

1. **Generality beyond Gaussians:** Does the information-geometric framework carry over to non-Gaussian transition kernels? If the theory is limited to the Gaussian case, how should we interpret the broader claims about geometry-aware sampling? A positive answer with concrete examples would strengthen the theoretical contribution.

2. **Constant informational progress:** Can you explain more precisely why maintaining "approximately constant increments in the informational distance" is beneficial? What goes wrong concretely when this condition is violated, beyond the general argument about discretization error?

3. **Statistical significance:** Can you provide confidence intervals or statistical tests for the experimental results? Given the small margins of improvement, this is essential to assess whether DVS offers a genuine advantage.

**Limitations:**

The authors do not sufficiently discuss the limitation that their method is only applied over selected portions of the trajectory, nor do they address the lack of statistical significance testing in the experiments.

**Strengths And Weaknesses:**

- **Originality:** The idea of viewing the diffusion sampling trajectory through the lens of information geometry and using the Fisher-Rao metric to drive adaptive step-size control is interesting and provides a principled geometric motivation.
- **Significance:** The method is training-free and can be integrated into existing diffusion frameworks (GRUM, GDSS) without modifying the model or its parameters.

- **Presentation (Section 3 and Appendix A):** The methodology section would benefit from a significantly clearer exposition. Appendix A is sketchy and lacks references. In particular, Equation (4) introducing the Riemannian line element $ds^2$ is not explained and this notion is used repeatedly throughout the paper.
- **Soundness -- Gaussian example (Section 3.2):** The pedagogical example based on Gaussian distributions leads to Equation (9), where $ds^2 = \frac{dt}{\sigma_t^2} \|df_t\|_2^2$. This result is somewhat disappointing: it essentially says the infinitesimal change should be rescaled by the variance. It raises the question of whether the full information-geometric machinery is necessary to arrive at this conclusion. More importantly, it is unclear whether the theory extends beyond Gaussian transition kernels.
- **Significance -- experimental evidence:** The reported improvements are very small. The authors should provide confidence intervals to establish whether the differences are statistically significant rather than falling within experimental variance.
- **Soundness -- partial application (Section 4.1):** The implementation detail that DVS is applied only over selected portions of the sampling trajectory raises doubts about the method's general usefulness. If the adaptive strategy needs to be restricted to specific intervals for stability, this is a significant practical limitation that should be discussed more openly.

---

> ### Author Rebuttal · Authors · 2026-03-31
>
> We sincerely appreciate your constructive feedback and insightful comments. Our point-by-point responses are provided below.
>
> ### **Response to Q1: Beyond Gaussian Kernels**
>
> Our method adapts the step size based on how rapidly the underlying distribution evolves along the sampling trajectory. Importantly, this principle depends only on the evolution of the probability distribution and is therefore independent of whether the dynamics are continuous or discrete, or whether the transition kernel is Gaussian. The Fisher–Rao metric provides a general way to quantify distributional change via the sensitivity of log-probability, and applies to any parameterizable family of transition distributions. In our paper, the Gaussian assumption is used only to obtain a closed-form expression for efficient computation.
>
> This generality can be illustrated by discrete diffusion models such as D3PM, where the state space is categorical and the trajectory ${p_t}$ lies on the probability simplex. In this setting, the Fisher–Rao metric becomes $ds^2 = \sum_i (dp_i)^2 / p_i = \mathbb{E}_{i \sim p}[(d \log p_i)^2]$, which depends only on the evolution of probabilities and not on any Gaussian structure. This example shows that our geometry-aware step-size adaptation operates at the level of probability geometry and naturally extends beyond Gaussian kernels.
>
> ### **Response to Q2: Constant Information Increments**
>
> The key idea of DVS is to maintain a uniform rate of distributional change along the sampling trajectory, that is, a constant informational speed on the statistical manifold. This is particularly beneficial because diffusion dynamics are highly non-uniform: some regions evolve rapidly while others change slowly. If this variation is ignored, large steps in rapidly changing regions can miss critical transitions, resulting in biased probability transport and accumulated discretization errors, while overly small steps in smooth regions lead to unnecessary computation without improving sample quality. By enforcing approximately constant increments in the Fisher–Rao distance, DVS ensures that each step induces a comparable change in the distribution, automatically allocating smaller steps to high-variation regions and larger steps to smoother ones, thereby balancing accuracy and efficiency.
>
> ### **Response to Q3: Statistical Significance**
>
> Using the GDSS model on the QM9 dataset, we compared the fixed-step Euler-Maruyama sampler with our DVS-driven Euler-Maruyama sampler across 10 independent runs using identical random seeds. We report the mean ± standard deviation for each metric, perform paired t-tests based on the seed-wise results, and calculate the 95% confidence intervals (CI) for the performance differences. The results are summarized in the table below. The "Diff." column denotes the absolute performance change of DVS compared to the Fixed-Step baseline. All three metrics show statistically significant improvements under a paired $t$-test, particularly a strong improvement in FCD, indicating stable and meaningful gains in distributional fidelity.
>
> | Metric | DVS (Ours) | Fixed-Step  | Diff. | 95% CI of Diff. | $p$-value | Significance |
> | :--- | :---: | :---: | :---: | :---: | :---: | :---: |
> | **Valid (%)** ↑ | $95.40 \pm 0.25$ | $95.18 \pm 0.19$ | $+0.21 \pm 0.28$ | $[0.01, 0.41]$ | $0.040$ | $p < 0.05$ |
> | **FCD** ↓ | $2.472 \pm 0.040$ | $2.576 \pm 0.044$ | $-0.104 \pm 0.017$ | $[-0.12, -0.09]$ | **< 0.001** | $p < 0.001$ |
> | **NSPDK** ↓ | $0.0033 \pm 0.0001$ | $0.0034 \pm 0.0001$ | $-0.0001 \pm 0.0001$ | $[-0.0001, -0.0000]$ | $0.011$ | $p < 0.05$ |
>
> ### **Response to Clarification on the Riemannian Line Element $ds^2$**
>
> We will clearly define and explain $ds^2$ in the revision. In our framework,  $ds^2$ is defined as the infinitesimal arc length under the Fisher-Rao metric, measuring the distance between neighboring transition distributions. Step sizes are then scaled adaptively according to the magnitude of distributional change, and by controlling the increment of $ds^2$, the solver maintains a uniform rate of progression in distribution space.
>
> ### **Response to Concern on Active Ranges**
>
> Active ranges can be viewed as a lightweight efficiency heuristic that allocates computation to regions with higher variation intensity. In early diffusion stages, high noise leads to a low signal-to-noise ratio where the drift is less informative and coarse steps are sufficient; as noise decreases, the drift becomes more informative and the trajectory exhibits sharper changes, making adaptive step-size control more beneficial. Restricting adaptation to these regions therefore improves efficiency without altering the DVS mechanism. Importantly, DVS itself supports full-trajectory adaptation without manual specification. We apply it over the entire interval $[0,1]$ on QM9 and ZINC250k, showing that the method is stable for fully automatic use, while active ranges remain an optional efficiency refinement.

---

> > ### Author Rebuttal · Reviewer_VfSh · 2026-04-03
> >
> > Thank you for your detailed responses, which should be incorporated into the final version of the paper.

---

### Decision · Program_Chairs · 2026-04-30

**Decision:**

Accept (regular)

**Comment:**

**Summary**

In this paper, the authors investigate and challenge a common assumption in diffusion models, namely the uniform time discretisation. While such uniform partition is easy to implement for practitioners it is likely sub-optimal. In this paper, the authors discuss a time discretisation based on the Fisher-Rao metrics. With this perspective, they introduce Drift Variation Score (DVS) which takes into account the instantaneous change in Fisher-Rao metrics. They apply their methodology to the case of graph generation where both the edges and the vertices are generated using a continuous diffusion models. In that case, they report improvements on molecular generation and social network graph generation baselines. While the improvements are minor they seem to hold on the various range of tasks and datasets.

**Reviewers concerns**

Reviewers mentioned the limited improvement as well as the motivation to apply their methodology to graph generation. The experimental concerns, especially regarding the statistical significance of the results, were resolved in the rebuttal period. I would recommend for the authors to emphasize their choice of focusing on graph generation in the revised version of the paper. Mainly, the following statement "Notably, it is particularly effective for graph generation, where the data is discrete and heterogeneous, and the dynamics involve abrupt structural changes such as connectivity or node/edge variations." should be emphasized in the main paper.